# SimGFM: Simplifying Discrete Flow Matching for Graph Generation

## Abstract

Discrete Flow Matching (DFM) presents a promising approach for graph generation; however, existing adaptations often introduce substantial complexity by incorporating task-specific heuristics, compromising the continuity equation and significantly expanding the hyperparameter space. Moreover, their sampling efficiency remains limited, as the required number of steps is often comparable to diffusion models, diminishing DFM's practical advantages. To address these limitations, we propose SimGFM, a simplified graph DFM for graph generation. SimGFM introduces a graph-structured rate formulation based on minimalist design principles—characterized by a clear mathematical expression, free of ad-hoc heuristics, consistent with the continuity equation; along with a targeted scheduler informed by our observation that, under uniform denoising, valid graph structures predominantly emerge near the end of the denoising trajectory. SimGFM achieves strong empirical results: on QM9, it matches prior models requiring 500–1000 steps with only 10 steps, and on most datasets, its performance at 50 steps matches or surpasses these baselines, demonstrating both efficiency and competitiveness.

## 1 Introduction

Graph generation is fundamental across domains from molecular chemistry to social networks, as graphs compactly represent complex relations and generate realistic structured data. Recent advances include continuous-time discrete diffusion frameworks (Xu et al., 2024; Siraudin et al., 2024) and discrete-flow frameworks (QIN et al., 2025; Campbell et al., 2024; Gat et al., 2024).

Diffusion models (Ho et al., 2020; Nichol & Dhariwal, 2021; Vignac et al., 2022) tightly couple training and sampling: once components such as the noise schedule or rate matrix are modified (Nichol & Dhariwal, 2021; Karras et al., 2022; Xu et al., 2024; Siraudin et al., 2024), retraining is typically required, incurring substantial computational cost. By contrast, discrete-flow models (Campbell et al., 2024; Gat et al., 2024) decouple training from sampling, allowing sampling adaptations without retraining and thus greater flexibility for diverse data distributions.

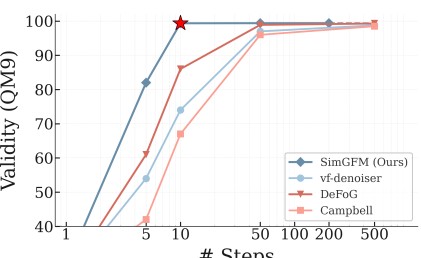

Figure 1: Validity on QM9 vs. sampling steps. Campbell (red) requires many steps, while vf-denoisers (blue) achieve higher validity with fewer steps. SimGFM further improves efficiency, reaching over 99% validity in 10 steps.

In CV/NLP, flow matching has markedly accelerated sampling, in some cases enabling near one-step generation (Song et al., 2023; Liu et al., 2022; Lee et al., 2024; Geng et al., 2025). However, in graph generation, existing discrete-flow models remain computationally costly and require nearly as many steps as diffusion-based approaches, leaving the potential sampling efficiency of flow matching largely unrealized (QIN et al., 2025; Hou et al., 2025).

As shown in Fig. 1, Campbell et al. (2024) derive a closed-form rate matrix (Eq. 5) from the posterior endpoint $p_{1|t}(\cdot \mid X_t)$, but its posterior expectations and combinatorial bookkeeping are costly for graphs. Building on this, recent SOTA model (QIN et al., 2025) augments the Campbell field with heuristic velocity terms to gain accuracy, at the cost of (i) potential violations of the continuity equation, and (ii) added methodological complexity. By contrast, the vf-denoiser (Gat et al.,

2024) offers a concise scheduler-based formulation (Eq. 6), avoids posterior expectations, and shows strong few-step performance, making it a simpler and more effective backbone for graph DFM.

Motivated by these observations, we propose SimGFM, a vf-denoiser-based method that strictly adheres to the standard DFM formulation without auxiliary modules. In particular, while the vf-denoiser is simple and flexible, it still suffers from compounding denoising errors, i.e., the accumulation and propagation of small prediction errors along the iterative denoising trajectory (Boget, 2025). To address this, we introduce the rvf-denoiser (Eq. 11), a sampling-based variant that selects a single candidate outcome at each step, numerically more stable in finite-precision implementation and help mitigate compounding denoising errors. In addition, motivated by our observation that under uniform denoising, valid graph structures predominantly emerge near the end of the trajectory, SimGFM incorporates a targeted scheduler that allocates more updates to this endpoint region to better align sampling with discrete flow dynamics. On QM9 (Wu et al., 2018), SimGFM achieves 99.5% validity in just 10 steps, and across most datasets, it reaches or approaches SOTA performance with 10–50 steps, representing an *order-of-magnitude reduction* compared with diffusion/flow baselines (typically 500–1000), while also lowering hyperparameter tuning burden.

## 2 PRELIMINARIES

### 2.1 DISCRETE FLOW MATCHING

In this section, we introduce the core concepts of Discrete Flow Matching (DFM) (Campbell et al., 2024; Gat et al., 2024). Unlike diffusion models, which learn a data distribution via iterative noising and denoising, the goal of DFM is to learn a deterministic *probability path* $p_t$ from a simple source distribution $p_0$ (e.g., a sequence composed of a "mask" symbol) to a target data distribution $p_1$. The core of the model is to train a neural network to predict the *velocity field* $u_t$ of this probability path, which guides how samples evolve with time $t \in [0, 1]$ from the source to the target.

To build this framework, we first define a **conditional probability path** from a specific source sample $x_0 \sim p_0$ to a specific target sample $x_1 \sim p_1$. A simple and effective choice is their convex combination:

$$p_t\big(x^i \mid x_0, x_1\big) = (1 - \kappa_t)\, \delta_{x_0^i}\big(x^i\big) + \kappa_t\, \delta_{x_1^i}\big(x^i\big), \tag{1}$$

where $x^i$ is the $i$-th element of the sequence, $\delta$ is the Dirac delta (point mass), and $\kappa_t$ is a schedule increasing monotonically from $\kappa_0 = 0$ to $\kappa_1 = 1$. This formula states that at time $t = 0$, the sample coincides with the source $x_0$, and at $t = 1$ it fully transforms into the target $x_1$.

To simulate generation along the prescribed path $p_t(x)$ for $t \in [0, 1]$, DFM adopts the **continuous-time Markov chain** (CTMC) paradigm: the sample $X_t$ makes jumps over a state space $\mathcal{D}$ as time $t$ evolves continuously on $[0, 1]$. DFM focuses on a model that predicts the **rate of change of probabilities** for each coordinate (token) of the current sample $X_t$ with $N$ tokens. Thus, for a sample $X_t \sim p_t$, each token updates independently as

$$X_{t+h}^i \sim \delta_{X_t^i}(\cdot) + h\, u_t^i(\cdot, X_t), \tag{2}$$

where $\delta_{X_t^i}$ denotes a Dirac mass at the current value and $u_t^i$ is the probability velocity field for the $i$-th coordinate. If the probabilistic velocity $u_t$ *generates* the probability path $p_t$, it means that for all $t \in [0, 1)$ and any sample $x_t \sim p_t$, updating each position $i$ using the rule above equation 2 yields $x_{t+h} \sim p_{t+h} + o(h)$.

Moreover, the velocity $u_t$ should satisfy the following **rate conditions**:

$$\sum_{x^i \in [K]} u_t^i(x^i, z) = 0, \qquad u_t^i(x^i, z) \ge 0 \quad \forall i \in [D],\ x^i \ne z^i. \tag{3}$$

Furthermore, prior work (Campbell et al., 2024; Gat et al., 2024) shows that a **continuity equation** (also called the Kolmogorov forward equation) holds in discrete flow matching, describing the time derivative of the state-marginal probability $\dot{p}_t(x)$, $x \in \mathcal{S}$:

$$\dot{p}_t(x) + \mathrm{div}_x(p_t\, u_t) = 0, \tag{4}$$

where $\mathrm{div}_x\big(p_t\, u_t\big) = \sum_{z \in \mathcal{S}} \sum_{i=1}^{D} \delta_x(z^{\bar{i}}) \Big[ p_t(x)\, u_t^i(x^i, x) - p_t(z)\, u_t^i(x^i, z) \Big]$, measures the total outflow (probability flow $x \to z$) minus total inflow ($z \to x$) at state $x \in \mathcal{S}$, and $\delta_x(z^{\bar{i}}) =$

$\prod_{j \neq i} \delta_{x^j}(z^j)$ indicates that only pairs $(x, z)$ agreeing on all coordinates except possibly the $i$-th are considered when computing the flow. Intuitively, the continuity equation expresses that the rate of change of probability mass at $x$ equals the net effect of the probability flow $p_t u_t$ at $x$. It has been shown that if the continuity equation holds, then $u_t$ can generate the probability path $p_t$.

The choice of $u_t$ is crucial. Two commonly used constructions for the rate matrix are:

**(1) Campbell's construction.** Campbell et al. (2024) provide a closed-form solution for the rate matrix $u_t$:

$$u_t^*(x, z | z_1) = \frac{\text{ReLU}\left[\partial_t p_{t|1}(x \mid z_1) - \partial_t p_{t|1}(z \mid z_1)\right]}{Z_t^{>0} p_{t|1}(z \mid z_1)}, x \neq z. \tag{5}$$

where $p_{t|1}(z \mid x)$ means the state $z$ at time $t$ given the state $x$ at time 1 and $Z_t^{>0} = \left|\{z_t : p_{t|1}(z_t \mid z_1) > 0\}\right|$, the diagonal case is set by $u_t^*(x, x | z_1) = -\sum_{x \neq z} u_t^*(x, z | z_1)$. Finally, the rate matrix is obtained by taking the posterior expectation: $u_t(x, z) = \mathbb{E}_{p_{1|t}(z_1|z)}[u_t^*(x, z \mid z_1)]$.

**(2) Vf-denoiser.** Gat et al. (2024) propose the vf-denoiser:

$$u_t^i(x^i, z) = \frac{\dot{\kappa}_t}{1 - \kappa_t}\left[p_{1|t}(x^i \mid z) - \delta_{z^i}(x^i)\right], \tag{6}$$

where $p_{1|t}(x \mid z)$ means the state $x$ at time 1 given the state $z$ at time $t$ and $\kappa_t$ is a scheduler (a monotone time mapping) satisfying $\dot{\kappa}_t \geq 0$, $\kappa_0 = 0$, and $\kappa_1 = 1$.

Both constructions depend on the prior $p_{1|t}(\cdot \mid z_t)$, which is typically estimated by a trained model; we denote the model output by $p_{1|t}^\theta(\cdot \mid z_t)$. The training objective is

$$\mathcal{L}(\theta) = -\sum_{i \in [N]} \mathbb{E}_{t,\,(X_0, X_1),\,X_t} \log p_{1|t}^\theta\left(X_1^i \mid X_t\right). \tag{7}$$

## 2.2 DISCRETE FLOW MATCHING ON GRAPHS

Applying the Discrete Flow Matching (DFM) framework to graph generation requires accounting for the unique structure of graphs—namely, sets of nodes and edges. We represent a graph with $N$ nodes as $G = (X, E)$, where $X = \{x^{(i)}\}_{i=1}^N$ is the set of node attributes and $E = \{e^{(ij)}\}_{1 \leq i < j \leq N}$ is the set of edge attributes. Based on Eq. 1, the probability path over graphs factorizes as

$$p_t(G_t \mid G_0, G_1) = \prod_{i=1}^N p_t\left(x_t^{(i)} \mid x_0^{(i)}, x_1^{(i)}\right) \prod_{1 \leq i < j \leq N} p_t\left(e_t^{(ij)} \mid e_0^{(ij)}, e_1^{(ij)}\right), \tag{8}$$

where $G_0 \sim p_0$ is a prior noise graph and $G_1 \sim p_1$ is a real data graph. Given this factorization, the sampling process for graphs follows the general update rule in Eq. 2: each node or edge is updated independently according to its velocity field,

$$G_{t+\Delta t}^{(k)} \sim \delta_{G_t^{(k)}}(\cdot) + \Delta t \cdot u_t^{(k)}(\cdot, G_t), \tag{9}$$

where $k$ denotes either a node index $(i)$ or an edge index $(ij)$. Iterating this process from $t = 0$ to $t = 1$ yields a generated graph.

### 2.2.1 EXISTING METHODS

Due to the structural complexity of graphs, graph generation is inherently more challenging than image or text generation. Although DFM has solid theoretical foundations, directly applying it to complex graph structures often yields suboptimal results. Consequently, researchers have developed a range of auxiliary or heuristic techniques to improve performance.

**Fine-tuning the model output.** This line of work optimizes the predictor $\phi_\theta$ to produce graphs with desired properties. For example, GGFLOW (Hou et al., 2025) adopts a two-stage strategy: first pretraining with standard flow-matching loss to learn $p_\theta(G_1 \mid G_t)$, and then fine-tuning via reinforcement learning (RL). Reward functions tied to graph properties (e.g., docking scores, connectivity) guide RL, yielding an optimized policy $p_\theta^{\text{RL}}(G_1 \mid G_t)$.

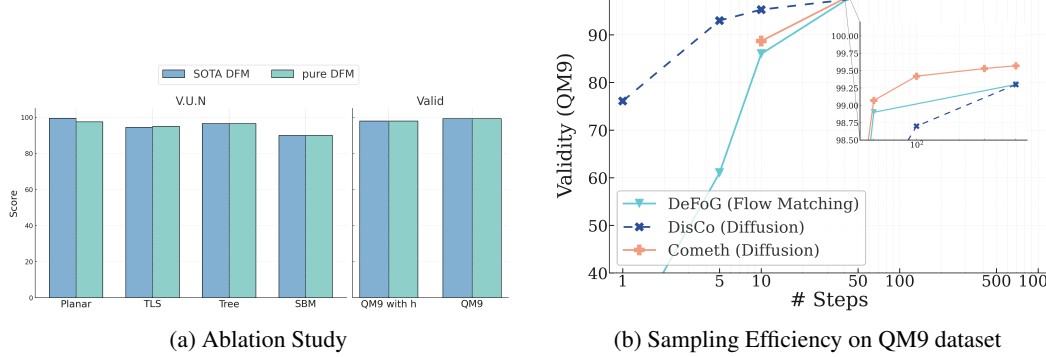

(a) Ablation Study    (b) Sampling Efficiency on QM9 dataset

Figure 2: Comparison of (a) ablation study and (b) sampling efficiency on the QM9 dataset.

**Modifying the velocity field.** Another line directly alters the sampling dynamics. DEFOG (QIN et al., 2025), for instance, augments Campbell's base field (Eq. 5) with heuristic terms:

$$u_t(\cdot \mid G_1) = u_t^*(\cdot \mid G_1) + \omega\, u_t^\omega(\cdot \mid G_1) + \eta\, u_t^{\mathrm{DB}}(\cdot \mid G_1), \tag{10}$$

where $u_t^*$ is the base velocity from Campbell's construction, $u_t^\omega$ is a target-guidance term weighted by $\omega$, and $u_t^{\mathrm{DB}}$ is a stochastic exploration term weighted by $\eta$.

### 2.2.2 OPEN CHALLENGES IN GRAPH DFM

**Violations of the continuity equation.** Directly fine-tuning the model output or modifying the velocity field (e.g., target-guidance heuristics) can break the core constraints required by Eq. 4, such as mass conservation and nonnegativity. In practice, these approaches often rely on auxiliary adjustments (e.g., normalization or clipping), which act as external interventions on the probability flow. While they may work empirically, such strategies lack a firm theoretical foundation and deviate from the standard DFM formulation.

**Methodological complexity.** Many enhancements to DFM introduce additional heuristics and design choices, which increase the overall modeling complexity and reduce reproducibility. These techniques expand the configuration space, making it harder to conduct systematic evaluation across tasks and datasets. As shown in Figure 2a, our experiments further indicate that, on some benchmarks, SOTA variant (QIN et al., 2025) do not consistently outperform pure baselines.

**Sampling efficiency.** In CV and NLP, flow models are valued for reducing the number of sampling steps compared to diffusion models. However, in graph generation, the steps required by current DFM methods remain comparable to those of diffusion approaches. This can be observed in Figure 2b, where existing graph DFM methods require nearly the same number of steps as diffusion-based models, suggesting that the efficiency advantage of DFM is not yet fully realized.

## 3 PROPOSED FRAMEWORK

We propose **SimGFM**, a minimalist framework for graph DFM that adheres strictly to the standard formulation without introducing auxiliary modules, thereby preserving fidelity to flow-matching theory. The overall pipeline is illustrated in Figure 3.

### 3.1 VELOCITY FIELD OF SIMGFM

In the DFM framework, the choice of the velocity field $u_t$ is central. Campbell's construction (Eq. 5), while theoretically sound, requires conditioning on fixed endpoints and averaging over posterior distributions, which incurs substantial computational overhead and hinders low-step generation. To alleviate this, we adopt the vf denoiser (Eq. 6) as our backbone, valued for its simplicity and scheduler-based flexibility.

However, in complex graph generation tasks such as MOSES (Polykovskiy et al., 2020) and TLS (Madeira et al., 2024), the vanilla vf-denoiser still exhibits compounding denoising errors (Bo-

Figure 3: Our Proposed SimGFM Framework.

get, 2025). Therefore, we introduce **rvf-denoiser (random vf-denoiser)**, a sampling-based variant of the vf-denoiser

Rather than using the full posterior distribution $p_{1|t}(\cdot \mid z)$, rvf-denoiser samples a single candidate $x_{1|t}{}^i \sim p_{1|t}(\cdot \mid z)$ and constructs the following velocity field:

$$u_t^{\mathbf{rvf},i}(x^i, z) = \frac{\dot{\kappa}_t}{1 - \kappa_t}\Big[\delta_{x_{1|t}{}^i}(x^i) - \delta_{z^i}(x^i)\Big]. \tag{11}$$

Our update rule uniformly depends on $p_{1|t}(\cdot \mid G_t)$, which can be interpreted as the expectation over all possible terminal graphs $G_1$ conditioned on the current state $G_t$. In practice, this distribution is approximated by the model, and we denote the resulting estimate as $p_{1|t}^\theta(\cdot \mid G_t)$. However, $p_{1|t}^\theta$ always contains statistical noise. The vf-denoiser directly applies the scaling factor $h\frac{\dot{\kappa}_t}{1-\kappa_t}$ to this noise, causing severe error amplification and systematic numerical drift. To address this, we adopt a stochastic strategy that first samples according to $p_{1|t}$ and then updates (rvf-denoiser). By employing sparse sampling, the rvf-denoiser precludes such amplification by decoupling the scaling factor from dense noise, thereby guaranteeing numerical stability; see Section A.6 for a rigorous analysis. When the data contain only a single dominant structure, the two updates are nearly aligned and incur negligible extra cost.

**Proposition 1** (Unbiasedness of rvf-denoiser). *The rvf-denoiser is an unbiased estimator of the vf-denoiser. Specifically, taking expectation over all possible candidate targets $x_{1|t}{}^i$, we have*

$$\mathbb{E}_{x_{1|t}{}^i|z}\big[u_t^{\mathbf{rvf},i}(x^i, z)\big] = u_t^i(x^i, z). \tag{12}$$

The proof is provided in Appendix A.1. It shows that rvf-denoiser behaves identically to vf-denoiser in expectation.

**Corollary 1** (Consistency with DFM Updates). *Due to its unbiasedness, the rvf-denoiser also satisfies the consistency requirement of DFM for one-step updates. Consequently, iterative sampling with rvf-denoiser simulates a distribution path that is consistent in expectation with the theoretical trajectory $p_t$, up to error $o(h)$.*

The proof is deferred to Appendix A.2. This corollary highlights that SimGFM is not a heuristic modification but a theoretically grounded alternative, fully consistent with the DFM framework in expectation. This expectation-level agreement ensures that the rvf-denoiser is a legitimate solver for DFM and that it exhibits improved numerical stability under finite-precision arithmetic.

**Proposition 2** (Variance Characterization of rvf-denoiser). *Conditioned on the current state $x_t$, the one-step update of the rvf-denoiser exhibits strictly higher variance than that of the vf-denoiser in the sense of positive semi-definite (PSD) matrices. Specifically:*

$$\mathrm{Var}\left(\delta_{x_t} + hu^{\mathbf{rvf}}\right) \succeq \mathrm{Var}\left(\delta_{x_t} + hu^{\mathbf{vf}}\right). \tag{13}$$

The detailed proof is provided in Appendix A.5. This inequality confirms that while both methods share the same expectation (consistency), the rvf-denoiser introduces structured stochasticity into the

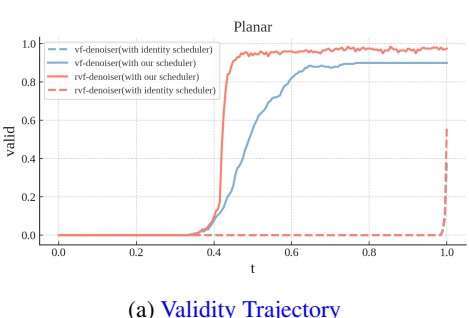
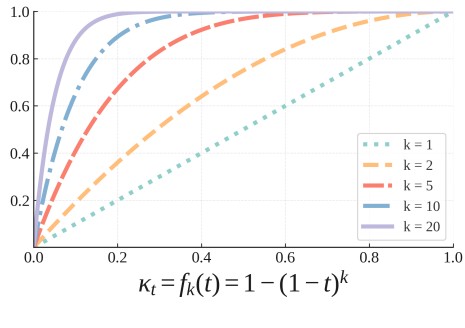

(a) Validity Trajectory

(b) Polynomial Scheduler Curves

Figure 4: (a) On Planar, the baseline (dashed) rises sharply only at the very end, suggesting that valid graphs emerge predominantly in the late denoising phase. Using $f_k(t)$ with $k = 10$, SimGFM (solid) allocates more steps to this late-stage refinement, improving validity. (b) The polynomial scheduler $f_k(t)$ flattens near $t = 1$ at higher $k$, concentrating steps in the critical refinement region.

generative trajectory. This stochastic trajectory effectively prevents the scale factor from amplifying model prediction errors, thereby improving numerical stability (see Appendix A.6).

## 3.2 THE CHOICE OF SCHEDULER

The temporal dynamics of discrete graph generation differ significantly from continuous domains. As illustrated in Figure 4a, we analyze the denoising trajectory on the Planar dataset and observe a critical phenomenon: under uniform denoising, valid graph structures emerge almost exclusively near the endpoint regime $t \rightarrow 1$. In the early and mid stages, structural validity remains close to zero, suggesting that a large portion of the computation budget is spent in regions that contribute little to the formation of valid structures.

This empirical pattern motivates the use of a non-uniform scheduler that allocates more updates near $t = 1$, where graph validity is most sensitive. Building upon the time-distortion strategy popularized by DeFoG (QIN et al., 2025), we adopt a polynomial scheduler of the form $f_k(t) = 1 - (1 - t)^k$ with $k \geq 1$ (Figure 4b). A larger $k$ slows down noise progression in the endpoint region, allowing the model to devote finer-grained updates precisely where valid structures are formed. As shown in Figure 4a, this leads to notably earlier and smoother emergence of valid graphs, in contrast to the sharp late-stage jump exhibited by baseline methods.

Importantly, while this scheduler substantially enhances the performance of our velocity-field-based formulation (Sec. 3.1), it is incompatible with Campbell's construction. Campbell's formulation was not derived with any scheduler in mind, and applying it with non-uniform schedules typically requires a time-distortion approximation. As analyzed in Appendix A.4, this approximation causes Campbell's updates to diminish rapidly under high-$k$ schedulers, leading to vanishing refinements in the endpoint region. In contrast, our vf/rvf velocity modeling maintains stable update magnitudes even under very high-order scheduling, enabling efficient targeted refinement and supporting high validity with significantly fewer denoising steps.

## 3.3 TRAINING AND SAMPLING PROCEDURES OF SIMGFM

Our framework follows the standard procedure of DFM (see Figure 3), but its core driving mechanism—the construction of the rate matrix—is redesigned to be more direct and efficient.

**Training.** We design the training procedure of **SimGFM** as shown in Algorithm 1. The entire objective centers on a single task: to teach a graph neural network $f_\theta$ to accurately predict the final, clean target graph $G_1$ from a halfway-evolved, ambiguous intermediate graph $G_t$. Each training iteration begins by sampling a real graph $G_1$ from the dataset and a time point $t$. An intermediate state $G_t$ between pure noise and real data is then generated according to **Eq. 8**. Next, the noised graph $G_t$, together with the current time $t$, is fed into the network to produce a prediction of the posterior distribution over the original graph $G_1$. Finally, we optimize the model parameters by

---

**Algorithm 1** SimGFM Training & Sampling

---

1 **Input:** Graph dataset $\mathcal{D} = \{G^1, \ldots, G^M\}$
2 **while** $f_\theta$ not converged **do**
3    Sample $G_1 \sim \mathcal{D}$
4    Sample $t \sim \mathcal{T}$
5    Sample $G_0 \sim p_0(G_0)$
6    Sample $G_t \sim p_t(G_t|G_0, G1)$   ▷ Noising
7    $p^\theta_{1|t}(\cdot|G_t) \leftarrow f_\theta(G_t, t)$   ▷ Denoising
8    loss $\leftarrow$ CE$_\lambda(G_1, p^\theta_{1|t}(\cdot|G_t))$
9    optimizer.step(loss)
10 **end while**

1 **Input:** # graphs to sample $S$
2 **for** $i = 1$ to $S$ **do**
3    Sample $N$ from train set        ▷ # Nodes
4    Sample $G_0 \sim p_0(G_0)$
5    **for** $t = 0$ to $1 - \Delta t$ with step $\Delta t$ **do**
6       $p^\theta_{1|t}(\cdot|G_t) \leftarrow f_\theta(G_t, t)$   ▷ Denoising prediction
7       $G^\theta_{1|t} \sim p^\theta_{1|t}(\cdot \mid G_t)$   ▷ Sample a potential graph
8       $u_t(\cdot, G_t) \leftarrow \frac{\dot{\kappa}_t}{1-\kappa_t}\left[\delta_{G^\theta_{1|t}}(\cdot) - \delta_{G_t}(\cdot)\right]$
9       $G_{t+\Delta t} \sim \delta_{G_t}(\cdot) + \Delta t \cdot u_t(\cdot, G_t)$ ▷ Update graph
10    **end for**
11    Store $G_1$
12 **end for**

---

computing the cross-entropy between this predicted distribution and the ground-truth graph.

$$\mathcal{L}(\theta) = - \sum_{i \in [N]} \mathbb{E}_{t, (G_0, G_1), G_t} \log p^\theta_{1|t}\big(x^i_1 \mid G_t\big) \; - \sum_{1 \leq i < j \leq N} \mathbb{E}_{t, (G_0, G_1), G_t} \log p^\theta_{1|t}\Big(e^{ij}_1 \mid G_t\Big)$$
(14)

Here, $x^i_1$ denotes the attribute of the $i$-th node in $G_1$ (i.e., the target label), and $e^{ij}_1$ denotes the attribute of the node pair $(i, j)$ in $G_1$: the value 1 indicates that the edge is absent, while any other value represents the attribute of an existing edge.

**Sampling.** The sampling process of SimGFM, shown in Algorithm 1, realizes graph generation as a direct evolution from chaos to order. It starts from a noise graph $G_0$ sampled entirely from the prior distribution. The model then iteratively evolves from $t = 0$ to $t = 1$ through a sequence of discrete time steps $\Delta t$. At each step $t$, given input $G_t$, the model predicts the posterior distribution $p^\theta_{1|t}(\cdot \mid G_t)$ of the final target graph. Updates are performed according to Eq. 9.

Rather than averaging over the full distribution, we **sample a concrete candidate target graph** $G^\theta_{1|t}$, which provides a sharp provisional direction for the current state. The rvf-denoiser then constructs a rate matrix $u_t$ that links $G_t$ only to this candidate target. The graph is updated via the corresponding Markov jump process, yielding $G_{t+\Delta t}$. Repeating this predict–sample–update cycle gradually transforms pure noise into a structured graph at $t = 1$ that matches the target distribution.

### 3.4 Permutation Invariance Guarantees

Graph generative models should respect the permutation symmetries of graphs: both training and sampling must be independent of node indices. In our model, we ensure: (1) the loss is permutation-invariant; (2) the backbone denoiser is permutation-equivariant; (3) the one-step update kernels of both *vf*- and *rvf*-denoisers are permutation-equivariant; (4) consequently, the overall training objective and the sampling distribution are permutation-invariant. Full proofs are in the Appendix A.3.

## 4 Experiments

### 4.1 Experimental Setup

**Datasets.** We evaluate SimGFM across three task groups: (1) *generic graph generation* — Planar, SBM (Martinkus et al., 2022), Tree (Bergmeister et al., 2023), Ego-small, Community-small, Grid (Jo et al., 2022); (2) *molecular graph generation* — QM9 / QM9-with-H (Wu et al., 2018), MOSES (Polykovskiy et al., 2020); and (3) *conditional generation* — TLS (Madeira et al., 2024). Following prior work, we adopt the standard evaluation protocol for each dataset, reporting Valid/Unique/Novel (V.U.N.), Ratio, Fréchet ChemNet Distance (FCD), and graph statistics distances (Degree-MMD, Clustering-MMD, Orbit-MMD).

**Baselines.** We compare against major families of graph generative models. **Autoregressive models** include GraphRNN (You et al., 2018), GRAN (Liao et al., 2019), GraphGen (Goyal et al., 2020)

Table 1: Graph generation performance on the synthetic datasets: Planar, Tree and SBM. V.U.N. denotes Valid, Unique, and Novel, with Ratio closer to 1 indicating better alignment. Values are mean ± std from five runs of 40 graphs each. Best and second-best results are in bold and underline.

| Model | Class | # Steps ↓ | Planar | | Tree | | SBM | |
|---|---|---|---|---|---|---|---|---|
| | | | V.U.N. ↑ | Ratio ↓ | V.U.N. ↑ | Ratio ↓ | V.U.N. ↑ | Ratio ↓ |
| Train set | — | — | 100 | 1.0 | 100 | 1.0 | 85.9 | 1.0 |
| GraphRNN | Autoregressive | — | 0.0 | 490.2 | 0.0 | 607.0 | 5.0 | 14.7 |
| GRAN | Autoregressive | — | 0.0 | 2.0 | 0.0 | 607.0 | 25.0 | 9.7 |
| BiGG | Autoregressive | — | 5.0 | 16.0 | 75.0 | 5.2 | 10.0 | 11.9 |
| GraphGen | Autoregressive | — | 7.5 | 210.3 | 95.0 | 33.2 | 5.0 | 48.8 |
| AUTOGRAPH | Autoregressive | — | 87.5 | 1.5 | — | — | **92.5** | 3.4 |
| EDGE | Diffusion | 1000 | 0.0 | 431.4 | 0.0 | 850.7 | 0.0 | 51.4 |
| BwR (EDP-GNN) | Diffusion | 1000 | 0.0 | 251.9 | 0.0 | 11.4 | 7.5 | 38.6 |
| DiGress | Diffusion | 1000 | 77.5 | 5.1 | 90.0 | 1.6 | 60.0 | 1.7 |
| HSpectre | Diffusion | — | 95.0 | 2.1 | **100.0** | 4.0 | 75.0 | 10.5 |
| GruM | Diffusion | — | 90.0 | 1.8 | — | — | 85.0 | **1.1** |
| DisCo | Diffusion | 500 | 83.6 | — | — | — | 66.2 | — |
| Cometh | Diffusion | 500 | 92.5 | — | — | — | 77.0 | — |
| Cometh-PC | Diffusion | — | 99.5 | — | — | — | — | — |
| CatFlow | Flow | — | 80.0 | — | — | — | 85.0 | — |
| DeFoG (50 steps) | Flow | **50** | 95.0 | 3.2 | 73.5 | 2.5 | 86.5 | 2.2 |
| DeFoG (1000 steps) | Flow | 1000 | 99.5 | **1.6** | 96.5 | 1.6 | 90.0 | 4.9 |
| SimGFM (20 steps) | Flow | **20** | 94.0±4.4 | 2.3±0.6 | 88.0±4.8 | 2.5±0.9 | 82.0±4.0 | 5.6±1.1 |
| SimGFM (50 steps) | Flow | 50 | 99.5±1.0 | 1.8±0.5 | 97.0±1.0 | 2.0±0.7 | 87.0±4.0 | 2.9±0.5 |
| SimGFM (200 steps) | Flow | 200 | **100.0**±0.0 | 9.3±2.6 | 99.5±1.0 | **1.5**±0.2 | 90.5±4.0 | 3.2±0.5 |

Table 2: Molecule generation on QM9. We present the results over five sampling runs of 10000 generated graphs each. We include the results of Relaxed Validity, which accounts for charged molecules, to facilitate comparison, as different methods may report varying types of validity.

| | Without Explicit Hydrogenes | | | | With Explicit Hydrogenes | | | | |
|---|---|---|---|---|---|---|---|---|---|
| Model | # Steps ↓ | Valid ↑ | Relaxed Valid ↑ | Unique ↑ | FCD ↓ | # Steps ↓ | Valid ↑ | Relaxed Valid ↑ | Unique ↑ | FCD ↓ |
| Training set | — | 99.3 | 99.5 | 99.2 | 0.03 | — | 97.8 | 98.9 | 99.9 | 0.01 |
| SPECTRE | — | 87.3 | — | 35.7 | — | — | — | — | — | — |
| GraphNVP | — | 83.1 | — | 99.2 | — | — | — | — | — | — |
| GDSS | — | 95.7 | — | 98.5 | 2.9 | — | — | — | — | — |
| DiGress | — | 99.0 | — | 96.2 | — | — | 95.4 | — | **97.6** | — |
| GruM | — | 99.2 | — | 96.7 | **0.11** | — | — | — | — | — |
| CatFlow | — | 99.8 | — | **100.0** | 0.44 | — | — | — | — | — |
| DisCo | — | 99.3 | — | — | — | — | — | — | — | — |
| Cometh | — | 99.6 | — | 96.8 | 0.25 | — | — | — | — | — |
| GRAPHARM | — | 90.25 | — | 95.62 | 1.22 | — | — | — | — | — |
| SID | — | 99.7 | — | — | 0.50 | — | — | — | — | — |
| CID | — | **99.9** | — | — | 1.76 | — | — | — | — | — |
| DeFoG (50 steps) | 50 | 98.9 | 99.2 | 96.2 | 0.26 | 50 | 97.1 | 98.1 | 94.8 | 0.31 |
| DeFoG (500 steps) | 500 | 99.3 | 99.4 | 96.3 | 0.12 | 500 | 98.0 | 98.8 | 96.7 | **0.05** |
| SimGFM (10 steps) | **10** | 99.5±0.0 | 99.7±0.0 | 95.0±0.2 | 0.92±0.0 | **10** | 93.7±0.2 | 95.6±0.3 | **97.6**±0.1 | 0.10±0.0 |
| SimGFM (50 steps) | 50 | 99.7±0.0 | **99.8**±0.0 | 96.3±0.0 | 0.13±0.0 | 50 | **98.4**±0.0 | **99.2**±0.1 | 97.1±0.1 | 0.10±0.0 |
| SimGFM (200 steps) | 200 | 99.8±0.0 | **99.8**±0.0 | 95.9±0.0 | 0.15±0.0 | 200 | **98.4**±0.1 | **99.2**±0.0 | 97.0±0.3 | 0.10±0.0 |

BiGG (Dai et al., 2020), and AUTOGRAPH (Chen et al., 2025). **GAN models** cover Graph-NVP (Madhawa et al., 2019) and SPECTRE (Martinkus et al., 2022). **Diffusion models** consist of DiGress (Vignac et al., 2022), GDSS (Jo et al., 2022), EDGE (Chen et al., 2023), BwR (Diamant et al., 2023), HSpectre (Bergmeister et al., 2023), GruM (Jo et al., 2023), DisCo (Xu et al., 2024), Cometh (Siraudin et al., 2024) and SID/CID (Boget, 2025) Finally, **Flow models** include DeFoG (QIN et al., 2025), CatFlow (Eijkelboom et al., 2024), and GGFlow (Hou et al., 2025).

Baseline results are from official implementations or reported numbers in the corresponding papers; further details in Appendix B.

## 4.2 OVERALL PERFORMANCE

> Requiring only **10–50 sampling steps**, SimGFM can match or even outperform state-of-the-art models across generic, molecular, and conditional graph generation tasks.

### 4.2.1 GENERIC GRAPH GENERATION

We evaluate SimGFM on the standard Planar, SBM, and Tree benchmarks. Table 1 reports two key metrics: (i) valid/unique/novel (V.U.N.) graphs and (ii) the Ratio of graph-statistic distances between generated and test sets relative to the train–test distance (lower is better). SimGFM demonstrates strong efficiency: on **Planar**, it achieves 99.5% V.U.N. with a Ratio of 1.8 using only **50** steps; on

Table 3: Generation results on the generic graph datasets. Results are the means of 3 different runs. The best results and the second-best results are marked **bold** and underline.

| Model | # Steps↓ | Ego-small | | | | Community-small | | | | Grid | | | |
|---|---|---|---|---|---|---|---|---|---|---|---|---|---|
| | | Deg.↓ | Clus.↓ | Orbit↓ | Avg.↓ | Deg.↓ | Clus.↓ | Orbit↓ | Avg.↓ | Deg.↓ | Clus.↓ | Orbit↓ | Avg.↓ |
| Training Set | - | 0.014 | 0.022 | 0.004 | 0.013 | 0.003 | 0.009 | 0.001 | 0.005 | 0.000 | 0.000 | 0.000 | 0.000 |
| GraphRNN | - | 0.090 | 0.220 | 0.003 | 0.104 | 0.080 | 0.120 | 0.040 | 0.080 | 0.064 | 0.043 | 0.021 | 0.043 |
| EDP-GNN | 1000 | 0.054 | 0.092 | 0.007 | 0.051 | 0.050 | 0.159 | 0.027 | 0.079 | 0.460 | 0.243 | 0.316 | 0.340 |
| GDSS | 1000 | 0.027 | 0.033 | 0.008 | 0.022 | 0.044 | 0.098 | 0.009 | 0.058 | 0.133 | 0.009 | 0.123 | 0.088 |
| DiGress | 500 | 0.028 | 0.046 | 0.008 | 0.027 | 0.032 | 0.047 | 0.009 | 0.025 | 0.037 | 0.046 | 0.069 | 0.051 |
| GGFlow | 500 | 0.005 | 0.033 | 0.004 | 0.014 | **0.011** | 0.030 | **0.002** | **0.014** | 0.030 | 0.000 | 0.016 | 0.015 |
| CatFlow | - | 0.013 | 0.024 | 0.008 | 0.015 | 0.018 | 0.086 | 0.007 | 0.037 | 0.115 | 0.004 | 0.075 | 0.065 |
| DeFoG (50 steps) | **50** | 0.034 | 0.012 | 0.067 | 0.039 | 0.029 | 0.157 | 0.052 | 0.079 | 0.004 | **0.000** | **0.000** | 0.001 |
| DeFoG (200 steps) | 200 | 0.056 | 0.149 | 0.068 | 0.091 | 0.022 | 0.040 | **0.002** | 0.022 | 0.001 | **0.000** | **0.000** | **0.000** |
| SimGFM (50 steps) | **50** | **0.004** | 0.024 | 0.006 | 0.011 | 0.038 | 0.081 | 0.008 | 0.043 | **0.000** | **0.000** | **0.000** | **0.000** |
| SimGFM (200 steps) | 200 | 0.006 | **0.009** | **0.001** | **0.005** | 0.031 | **0.027** | **0.002** | 0.020 | **0.000** | **0.000** | **0.000** | **0.000** |

Table 4: Large molecule generation performance. Only iterative denoising-based methods are reported here.

| | MOSES | | | | | | |
|---|---|---|---|---|---|---|---|
| Model | Val. ↑ | Unique. ↑ | Novelty ↑ | Filters ↑ | FCD ↓ | SNN ↑ | Scaf ↑ |
| Training set | 100.0 | 100.0 | 0.0 | 100.0 | 0.01 | 0.64 | 99.1 |
| AUTOGRAPH | 87.4 | **100.0** | 85.9 | 98.6 | 0.91 | **0.55** | — |
| DiGress | 85.7 | **100.0** | 95.0 | 97.1 | 1.19 | 0.52 | 14.8 |
| DisCo | 88.3 | **100.0** | 97.7 | 95.6 | 1.44 | 0.50 | 15.1 |
| Cometh | 90.5 | 99.9 | 92.6 | **99.1** | 1.27 | 0.54 | 16.0 |
| DeFoG (50 steps) | 83.9 | 99.9 | 96.9 | 96.5 | 1.87 | 0.50 | **23.5** |
| DeFoG (500 steps) | **92.8** | 99.9 | 92.1 | 98.9 | 1.95 | **0.55** | 14.4 |
| SimGFM (50 steps) | 88.7 | **100.0** | 95.9 | 98.4 | 0.39 | — | — |
| SimGFM (200 steps) | 90.8 | **100.0** | 94.8 | 99.0 | **0.29** | — | — |

Table 5: TLS conditional generation results.

| Model | TLS Dataset | |
|---|---|---|
| | V.U.N. ↑ | TLS Val. ↑ |
| Train set | 0.0 | 100 |
| GraphGen | 40.2 | 25.1 |
| BiGG | 0.6 | 16.7 |
| SPECTRE | 7.9 | 25.3 |
| DiGress | 13.2 | 12.6 |
| ConStruct | **99.1** | 92.1 |
| DeFoG (50 steps) | 44.5 | 93.0 |
| DeFoG (1000 steps) | 94.5 | 95.8 |
| SimGFM (50 steps) | 81.3 | 91.3 |
| SimGFM (200 steps) | 96.3 | **96.3** |

**Tree**, it reaches 99.5% V.U.N. and 1.5 Ratio at **200** steps; and on **SBM**, it matches the performance of DeFoG with **200** steps, compared to DeFoG's **1000**. These results highlight that a minimalist, well-founded design can deliver both competitiveness and efficiency.

We further assess structural fidelity on Ego-small, Community-small, and Grid. Table 3 shows that SimGFM with **200** steps achieves consistently small deviations across degree, clustering, and orbit statistics, reaching or approaching the best overall scores among all compared methods.

### 4.2.2 MOLECULAR GRAPH GENERATION

We further evaluate SimGFM on three molecular benchmarks. On **QM9**, Table 2 shows that SimGFM achieves SOTA performance at **200** steps, while already reaching 99.5% validity with only **10** steps, which is an order of magnitude fewer than the ∼ 500 steps typically required by diffusion models, thereby demonstrating substantial gains in sampling efficiency. On **QM9-with-H**, results in Table 2 indicate that SimGFM at **200** steps matches or surpasses the best reported scores across all metrics, and at just **50** steps achieves a FCD of 0.10. For the large-molecule dataset **MOSES**, Table 4 shows that SimGFM with **200** steps reduces FCD to 0.29, the lowest among all compared methods, while maintaining strong validity and uniqueness.

### 4.2.3 CONDITIONAL GENERATION

We evaluate conditional generation on TLS dataset. Performance is assessed by (i) TLS Valid, measuring consistency between generated graphs and provided labels, and (ii) V.U.N. (validity, uniqueness, and novelty), where a graph is considered valid if it is both planar and connected. For fairness, we report the mean performance of existing methods on two sub-

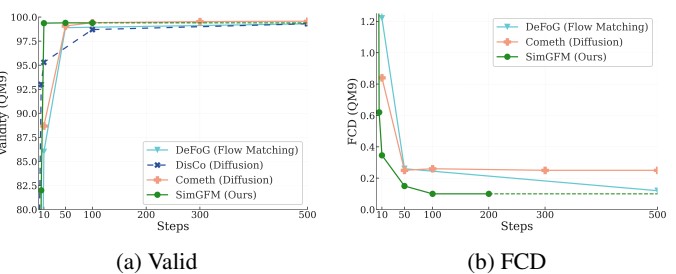

(a) Valid          (b) FCD

Figure 5: Sampling Efficiency on QM9

sets, as summarized in Table 5. SimGFM achieves **96.3**% TLS Valid and **96.3**% V.U.N. with only **200** steps, matching or surpassing DeFoG while requiring far fewer inference steps.

### 4.3 SAMPLING EFFICIENCY

We report validity and FCD as functions of sampling steps on QM9 (Figs. 5a and 5b). SimGFM surpasses 0.99 validity with only **10** steps, whereas other methods typically require at least **50**. This advantage arises from the DFM mechanism: by following a straighter probability path, SimGFM reaches high validity with substantially fewer refinement steps.

In terms of FCD (Fig. 5b), SimGFM decays rapidly from **0.92** at 10 to **0.15** at 200. Thus, **10** steps already attain performance once associated with **500 − 1000**, while **200** steps achieve best-in-class results, underscoring a significant improvement in sampling efficiency.

### 4.4 ABLATION STUDY

We study two training–sampling sensitive components: (i) the rate-matrix estimator (vf-denoiser vs. rvf-denoiser) and (ii) the DFM time scheduler $\kappa_t$.

Table 6 summarizes the effect of replacing the vf-denoiser with the rvf-denoiser under 200 sampling steps. On TLS conditional generation, rvf-denoiser improves Valid by an absolute 93.75 points ($2.50 \rightarrow 96.25$), indicating a substantial robustness gain under conditional constraints. On MOSES, Valid rises from $85.78$ to $89.39$ ($+3.61$). Overall, rvf-denoiser outperforms vf-denoiser across benchmarks and is a stronger default choice.

| Dataset | Vf-denoiser | Rvf-denoiser | Gain |
|---|---|---|---|
| TLS | 2.50 | 96.25 | +93.75 |
| QM9-with-H | 97.25 | 98.40 | +1.15 |
| MOSES | 85.78 | 89.39 | +3.61 |

Table 6: Rate-matrix ablation.

We further analyze the time scheduler $\kappa_t$. Results across datasets show that stronger front loading (larger $k$) benefits small step budgets, while moderate front loading ($5 \leq k \leq 10$) is more effective for larger budgets. Detailed results are provided in Appendix C (Table 9 and 10).

## 5 RELATED WORK

**Diffusion models** (Ho et al., 2020) treat generation as iterative denoising. Discrete variants like DiGress (Vignac et al., 2022) edit nodes and edges categorically while preserving marginals, achieving strong results on molecular and non-molecular datasets. Extensions such as EDGE (Chen et al., 2023), and DisCo (Xu et al., 2024) improve efficiency or structural modeling through mixture strategies, bandwidth constraints, or richer encodings. SID (Boget, 2025) partially mitigates compounding denoising errors by assuming conditional independence between intermediate states. Continuous-time variants (Campbell et al., 2022; Xu et al., 2024) employ CTMCs; e.g., Cometh (Siraudin et al., 2024) integrates random-walk features to boost validity, uniqueness, and novelty. Despite these advances, diffusion remains hindered by slow sampling and broader error accumulation.

**Flow Matching (FM)** offers a more efficient refinement paradigm, transporting noise to data via ODEs or CTMCs with improved stability (Lipman et al., 2022; Liu et al., 2022) and demonstrated success in vision domains (Esser et al., 2024; Ma et al., 2024). Its discrete extension, **DFM** (Campbell et al., 2024; Gat et al., 2024), extends the framework to categorical data, including graphs, by employing linear interpolation and CTMC dynamics. Subsequent works such as CatFlow (Hou et al., 2025), DeFoG (QIN et al., 2025), and GGFlow (Hou et al., 2025) enhance performance but rely on costly optimization, heuristics, or reinforcement learning, complicating the framework.

## 6 CONCLUSION

We presented SimGFM, a minimal yet strong framework for discrete flow matching on graphs. Our approach employs a clean CTMC formulation, a simple monotone scheduler, and the unbiased rvf-denoiser, which together are sufficient to match or surpass more complex systems using only 10–50 steps. These results demonstrate that principled probabilistic design choices, free from ad-hoc heuristics, can substantially improve sampling efficiency while maintaining strong performance.

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

# A PROOF

## A.1 PROOF OF UNBIASEDNESS OF RVF-DENOISER

**Proposition 1** (Unbiasedness of rvf-denoiser). *The rvf-denoiser is an unbiased estimator of the vf-denoiser. Specifically, taking expectation over all possible candidate targets $x^i_{1|t}$, we have*

$$\mathbb{E}_{x^i_{1|t}|z}\big[u^{\mathbf{rvf}}_t(x^i, z)\big] = u^i_t(x^i, z). \tag{15}$$

*Proof.* The derivation follows directly. Starting from the definition:

$$\mathbb{E}_{x^i_{1|t}|X_t}\big[u^{\mathbf{rvf}}_t\big] = \mathbb{E}_{x^i_{1|t}|X_t}\left[\frac{\dot{\kappa}_t}{1-\kappa_t}\Big(\delta_{x^i_{1|t}}(x^i) - \delta_{X^i_t}(x^i)\Big)\right]. \tag{16}$$

Moving constants outside of the expectation:

$$= \frac{\dot{\kappa}_t}{1-\kappa_t}\Big(\mathbb{E}_{x^i_{1|t}|X_t}[\delta_{x^i_{1|t}}(x^i)] - \delta_{X^i_t}(x^i)\Big). \tag{17}$$

By definition of expectation, $\mathbb{E}_{x^i_{1|t}|X_t}[\delta_{x^i_{1|t}}(x^i)]$ equals $p_{1|t}(x^i \mid X_t)$. Substituting, we obtain:

$$= \frac{\dot{\kappa}_t}{1-\kappa_t}\Big[p_{1|t}(x^i \mid X_t) - \delta_{X^i_t}(x^i)\Big] = u^i_t(x^i, X_t). \tag{18}$$

$\square$

## A.2 PROOF OF CONSISTENCY OF SIMGFM UPDATES WITH DFM

**Corollary 1** (Consistency with DFM Updates). *Due to its unbiasedness, the rvf-denoiser also satisfies the consistency requirement of DFM for one-step updates. Consequently, iterative sampling with rvf-denoiser simulates a distribution path that is consistent in expectation with the theoretical trajectory $p_t$, up to error $o(h)$.*

*Proof.* The validity of DFM relies on ensuring that each update approximately pushes the sample distribution from $p_t$ to $p_{t+h}$ in expectation. Itai et al. proved that vf-denoiser satisfies:

$$\mathbb{E}_{X_t}\left[\delta_{X_t}(x) + h\sum_{i=1}^N \delta_{X_t}(x^{\bar{i}})\, u^i_t(x^i, X_t)\right] = p_{t+h}(x) + o(h). \tag{19}$$

Applying the law of total expectation and the unbiasedness property, we obtain:

$$\mathbb{E}_{X_t, X_{1|t}}\left[\delta_{X_t}(x) + h\sum_{i=1}^N \delta_{X_t}(x^{\bar{i}})\, u^{\mathbf{rvf}}_t(x^i, X_t)\right] \tag{20}$$

$$= \mathbb{E}_{X_t}\left[\mathbb{E}_{X_{1|t}|X_t}\left[\delta_{X_t}(x) + h\sum_{i=1}^N \delta_{X_t}(x^{\bar{i}})\, u^{\mathbf{rvf}}_t(x^i, X_t)\right]\right] \tag{21}$$

$$= \mathbb{E}_{X_t}\left[\delta_{X_t}(x) + h\sum_{i=1}^N \delta_{X_t}(x^{\bar{i}}) \underbrace{\mathbb{E}_{X_{1|t}^i|X_t}\big[u^{\mathbf{rvf}}_t(x^i, X_t)\big]}_{=u^i_t(x^i, X_t)}\right] \tag{22}$$

$$= p_{t+h}(x) + o(h). \tag{23}$$

$\square$

### A.3 PROOF OF PERMUTATION INVARIANCE FOR RVF-DENOISER

#### A.3.1 NOTATION AND SETUP

We denote an undirected graph by $G = (x_{1:N}, e_{1 \leq i < j \leq N})$, where node variables take values in $\mathcal{X}$ and edge variables in $\mathcal{E}$. For any node permutation with matrix $P \in \{0,1\}^{N \times N}$, define the relabeling action on graph-indexed tensors by

$$\pi_P(X) = PX, \qquad \pi_P(A) = PAP^\top, \qquad [\pi_P(E)]_{\{i,j\}} = E_{\{P^{-1}(i), P^{-1}(j)\}}.$$

Let $\delta_G$ denote the Dirac measure at $G$, and $(\pi_P)_{\#}\mu$ the pushforward of a measure $\mu$ via $\pi_P$. Scalars such as $t, \Delta t, \kappa_t, \dot{\kappa}_t$ are invariant under $\pi_P$. We write $G_t$ for a noisy state, $f_\theta$ for the denoiser, and $p_{1|t}^\theta(\cdot \mid G_t)$ for the predicted clean-graph distribution.

#### A.3.2 BACKBONE EQUIVARIANCE

**Proposition 2.** *The attention-based Graph Transformer denoiser satisfies*

$$f_\theta(\pi_P(G_t), t) = \pi_P\big(f_\theta(G_t, t)\big), \qquad p_{1|t}^\theta(\cdot \mid \pi_P(G_t)) = \pi_P\big(p_{1|t}^\theta(\cdot \mid G_t)\big).$$

*Proof.* With shared projections $Q = XW_Q$, $K = XW_K$, $V = XW_V$, relabeling yields $Q' = PQ$, $K' = PK$, $V' = PV$. Shared edge bias/mask obeys $B' = PBP^\top$, $M' = PMP^\top$. The score matrix satisfies $L' = \frac{Q'K'^\top}{\sqrt{d_k}} + B' + M' = PLP^\top$. Row-softmax commutes with row permutations, hence $\text{Att}' = P\,\text{Att}\,P^\top$. Aggregation gives $Y' = \text{Att}'V' = P(\text{Att}V) = PY$. Pointwise residuals, layer normalizations, and MLPs commute with $P$. Multi-head attention and stacking preserve equivariance. $\square$

#### A.3.3 LOSS INVARIANCE

**Proposition 3.** *The training loss*

$$\mathcal{L}(\theta; G_t, G_1) = - \sum_{i \in [N]} \log p_{1|t}^\theta(x_1^i \mid G_t) \; - \sum_{1 \leq i < j \leq N} \log p_{1|t}^\theta(e_1^{ij} \mid G_t)$$

*is permutation-invariant:*

$$\mathcal{L}(\theta; \pi_P(G_t), \pi_P(G_1)) = \mathcal{L}(\theta; G_t, G_1).$$

*Proof.* By backbone equivariance, $p_{1|t}^\theta(\cdot \mid \pi_P(G_t)) = \pi_P\big(p_{1|t}^\theta(\cdot \mid G_t)\big)$. The node sum reindexes via $i \mapsto P(i)$; the unordered edge sum reindexes via $\{i, j\} \mapsto \{P(i), P(j)\}$. Reindexing does not change the sums, proving invariance. $\square$

#### A.3.4 ONE-STEP KERNEL EQUIVARIANCE

Define the vector fields and one-step kernels (with global scalars $\kappa_t, \dot{\kappa}_t, \Delta t$):

$$\widehat{G} \sim p_{1|t}^\theta(\cdot \mid G_t), \qquad u_t^{\text{rvf}}(\cdot, G_t) = \frac{\dot{\kappa}_t}{1 - \kappa_t}\big[\delta_{\widehat{G}}(\cdot) - \delta_{G_t}(\cdot)\big], \qquad K_t^{\text{rvf}} = \delta_{G_t} + \Delta t\, u_t^{\text{rvf}},$$

$$u_t^{\text{vf}}(\cdot, G_t) = \frac{\dot{\kappa}_t}{1 - \kappa_t}\Big[\mathbb{E}_{\widehat{G} \sim p_{1|t}^\theta(\cdot \mid G_t)} \delta_{\widehat{G}}(\cdot) - \delta_{G_t}(\cdot)\Big], \qquad K_t^{\text{vf}} = \delta_{G_t} + \Delta t\, u_t^{\text{vf}}.$$

**Proposition 4.** *Vf-denoiser. For any measurable set $\mathcal{S}$,*

$$K_t^{\text{vf}}(\pi_P(G_t), \pi_P(\mathcal{S})) = K_t^{\text{vf}}(G_t, \mathcal{S}).$$

*Proof.* From backbone equivariance, $\pi_P(\widehat{G}) \stackrel{d}{=} \widehat{G}' \sim p_{1|t}^\theta(\cdot \mid \pi_P(G_t))$. Pushforward gives $(\pi_P)_{\#}\mathbb{E}[\delta_{\widehat{G}}] = \mathbb{E}[\delta_{\pi_P(\widehat{G})}]$ and $(\pi_P)_{\#}\delta_{G_t} = \delta_{\pi_P(G_t)}$, hence $K_t^{\text{vf}}(\pi_P(G_t), \cdot) = (\pi_P)_{\#}K_t^{\text{vf}}(G_t, \cdot)$. $\square$

**Proposition 5.** *Rvf-denoiser. For any measurable set $\mathcal{S}$,*

$$K_t^{\mathrm{rvf}}(\pi_P(G_t), \pi_P(\mathcal{S})) = K_t^{\mathrm{rvf}}(G_t, \mathcal{S}) \quad \text{in distribution.}$$

*Proof.* With $\pi_P(\widehat{G}) \overset{d}{=} \widehat{G}' \sim p_{1|t}^\theta(\cdot \mid \pi_P(G_t))$ and $(\pi_P)_\# \delta_{\widehat{G}} = \delta_{\pi_P(\widehat{G})}$, the claim follows immediately. $\square$

### A.3.5 SAMPLING TRAJECTORY AND TRAINING OBJECTIVE

**Sampling invariance.** If the initial distribution $p_0$ and the noising kernel $p_t(G_t \mid G_0, G_1)$ are compatible with permutations (relabeling only changes indices, not structural dependence), then kernel equivariance implies, by the Markov property and induction over time steps, that for any time grid and finite set of times,

$$\Pr\big(G_{t_1} \in \mathcal{S}_1, \ldots, G_{t_k} \in \mathcal{S}_k\big) = \Pr\big(\pi_P(G_{t_1}) \in \mathcal{S}_1, \ldots, \pi_P(G_{t_k}) \in \mathcal{S}_k\big),$$

so the terminal sampling distribution over isomorphism classes is permutation-invariant.

**Training invariance.** Taking expectation over $(t, (G_0, G_1), G_t)$ in the loss shows that the overall training objective is permutation-invariant; the expected gradient is unchanged under node relabeling.

### A.4 THEORETICAL ANALYSIS OF SCHEDULER COMPATIBILITY AND UPDATE DYNAMICS

In this section, we provide the theoretical motivation for our choice of scheduler and strictly analyze the numerical behavior of different discrete flow matching formulations near the terminal time $t \to 1$.

### A.4.1 MOTIVATION: THE NECESSITY OF NON-LINEAR SCHEDULERS

Empirical observations on discrete graph generation (as discussed in Method) reveal a critical dynamical property: valid graph structures typically emerge only when the diffusion time $t$ is very close to 1. Consequently, a linear scheduler often wastes computational budget on early noisy stages. To address this, we employ a polynomial scheduler of the form:

$$f_k(t) = 1 - (1 - t)^k, \quad k \geq 1. \tag{24}$$

Let $\kappa_t = f_k(t)$. A larger $k$ (e.g., $k = 10$ or $20$) flattens the trajectory near $t = 1$, effectively increasing the sampling resolution in the region where structural validity is determined.

### A.4.2 INCOMPATIBILITY OF TIME-DISTORTION APPROXIMATIONS (CAMPBELL'S FORMULATION)

Campbell et al. (2024) proposed a discrete flow matching update based on *time distortion*. We prove here that this approximation suffers from vanishing updates when combined with the necessary high-$k$ schedulers derived above.

The inference process under time distortion approximates the flow by adjusting the time step magnitude based on $\kappa_t$. The update rule implies a transition proportional to the change in noise level:

$$x_{\kappa_{t+h}} \sim x_{\kappa_t} + (\kappa_{t+h} - \kappa_t) \cdot R, \tag{25}$$

where $R$ represents the rate or update direction. To analyze the behavior as $h \to 0$, we perform a Taylor expansion of the scheduler $f_k(t)$ around $t$:

$$\kappa_{t+h} = f_k(t + h) = f_k(t) + f_k'(t)h + O(h^2). \tag{26}$$

Substituting the derivative $f_k'(t) = k(1 - t)^{k-1}$, the effective update magnitude becomes:

$$\Delta\kappa \approx \kappa_{t+h} - \kappa_t = k(1 - t)^{k-1}h. \tag{27}$$

**Analysis as $t \to 1$:** When utilizing a scheduler with a large $k$ to improve validity, the term $(1 - t)^{k-1}$ approaches zero extremely rapidly as $t \to 1$. Consequently, the update probability mass $\Delta\kappa$ vanishes. This causes the sampling trajectory to "freeze" prematurely—the model fails to execute necessary structural refinements in the final steps because the effective step size under time distortion becomes numerically negligible.

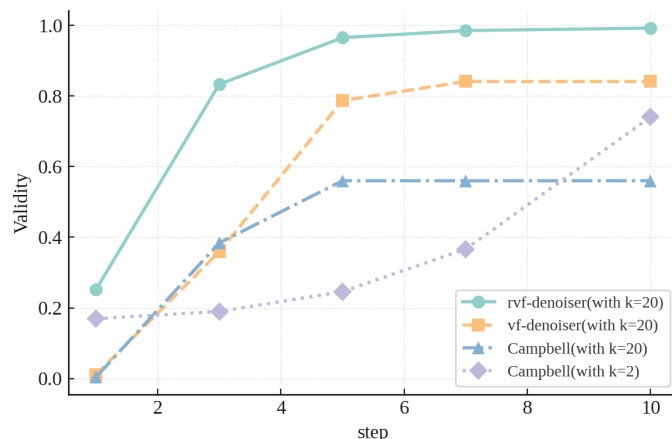

Figure 6: Comparison of validity trajectories on QM9 (10 steps, $k = 20$). While SimGFM (rvf/vf) continues to refine graph structures near $t \to 1$, the time-distortion baseline (Campbell) plateaus due to vanishing updates, validating our theoretical analysis.

### A.4.3    ROBUSTNESS OF THE VELOCITY-FIELD FORMULATION (SIMGFM)

In contrast, our proposed method (SimGFM) directly models the velocity field. The solver update rule for both rvf and vf is governed by the ratio of the rate of change to the remaining noise budget:

$$x_{t+h} \sim \delta_{x_t}(\cdot) + h \frac{\dot{\kappa}_t}{1 - \kappa_t} u, \tag{28}$$

where $u$ is the conditional vector field. Substituting the definitions for the polynomial scheduler $\kappa_t = 1 - (1 - t)^k$:

- The numerator: $\dot{\kappa}_t = k(1 - t)^{k-1}$.
- The denominator: $1 - \kappa_t = (1 - t)^k$.

The update coefficient simplifies to:

$$\frac{\dot{\kappa}_t}{1 - \kappa_t} = \frac{k(1 - t)^{k-1}}{(1 - t)^k} = \frac{k}{1 - t}. \tag{29}$$

Thus, the effective update rule behaves as:

$$x_{t+h} \sim \delta_{x_t}(\cdot) + h \frac{k}{1 - t} u. \tag{30}$$

**Conclusion:** Unlike the time-distortion formulation, the coefficient $h \frac{k}{1-t}$ does not vanish as $t \to 1$; instead, it compensates for the shrinking time horizon. This ensures that even with large $k$ values, the model maintains a significant probability of updating the graph structure up until the very end of the generation process. This theoretical derivation aligns with the experimental results on QM9, where SimGFM (rvf/vf) continues to improve validity in later steps, while the time-distortion baseline plateaus.

Figure 6 provides visual confirmation of this theoretical analysis. We conducted a controlled experiment on the QM9 dataset using a constrained budget of 10 steps with a high scheduler curvature ($k = 20$). The experimental curves clearly demonstrate the divergence in behavior near the terminal phase:

- **Campbell (Blue Line):** The validity curve flattens significantly as the step count progresses, confirming that the update magnitude $\kappa_{t+h} - \kappa_t$ becomes negligible, preventing the model from making final structural corrections.
- **SimGFM (Teal/Orange Lines):** Both the rvf and vf solvers maintain an upward trend in validity throughout the entire generation process. The non-vanishing coefficient $h \frac{k}{1-t}$ ensures that the model remains active and effective even as $t \to 1$, leading to superior final performance.

### A.5 VARIANCE ANALYSIS: DISTINGUISHING RVF-DENOISER FROM VF-DENOISER

We make precise that *given the current state*, vf and rvf share the same conditional mean but differ in conditional variance. By the law of total variance, the unconditional variance of rvf is therefore larger than or equal to that of vf, with strict inequality whenever the model is uncertain and the step size (scale) is nonzero.

#### A.5.1 NOTATION AND SETUP

Let $\mathbf{e}_t = \delta_{x_t}$ be the one-hot encoding of the current state $x_t$. Let $\mathbf{p} = p_{1|t}(\cdot \mid x_t)$ denote the model-predicted simplex probability at time $t$. Define the step scale $\lambda_t := h\,\dot{\kappa}_t/(1 - \kappa_t)$. For any random vector $\mathbf{X}$, we use the matrix-valued variance $\mathrm{Var}(\mathbf{X}) = \mathbb{E}\big[(\mathbf{X} - \mathbb{E}[\mathbf{X}])(\mathbf{X} - \mathbb{E}[\mathbf{X}])^\top\big]$. Let $\mathcal{G}_t := \sigma(x_t, \mathbf{p}, \lambda_t)$ denote the $\sigma$-field describing all randomness at time $t$ that is "current-information–measurable".

#### A.5.2 VF-DENOISER (DETERMINISTIC GIVEN $\mathcal{G}_t$)

$$u^{\mathrm{vf}} = \frac{\lambda_t}{h}\,(\mathbf{p} - \mathbf{e}_t), \qquad \mathbf{v}_{\mathrm{vf}} = \mathbf{e}_t + h\,u^{\mathrm{vf}} = (1 - \lambda_t)\mathbf{e}_t + \lambda_t\mathbf{p}. \tag{31}$$

Because $\mathbf{v}_{\mathrm{vf}}$ is a deterministic function of $\mathcal{G}_t$, its *conditional* variance vanishes:

$$\mathrm{Var}(\mathbf{v}_{\mathrm{vf}} \mid \mathcal{G}_t) = \mathbf{0}. \tag{32}$$

However, its *unconditional* variance generally does not vanish, since $(x_t, \mathbf{p}, \lambda_t)$ vary across trajectories:

$$\mathrm{Var}(\mathbf{v}_{\mathrm{vf}}) = \underbrace{\mathbb{E}[\mathrm{Var}(\mathbf{v}_{\mathrm{vf}} \mid \mathcal{G}_t)]}_{= \mathbf{0}} + \mathrm{Var}(\mathbb{E}[\mathbf{v}_{\mathrm{vf}} \mid \mathcal{G}_t]) = \mathrm{Var}((1 - \lambda_t)\mathbf{e}_t + \lambda_t\mathbf{p}). \tag{33}$$

#### A.5.3 RVF-DENOISER (STOCHASTIC GIVEN $\mathcal{G}_t$)

Draw a one-hot sample $\mathbf{S} \sim \mathrm{Cat}(\mathbf{p})$ conditionally on $\mathcal{G}_t$. Then

$$u^{\mathrm{rvf}} = \frac{\lambda_t}{h}\,(\mathbf{S} - \mathbf{e}_t), \qquad \mathbf{v}_{\mathrm{rvf}} = \mathbf{e}_t + h\,u^{\mathrm{rvf}} = (1 - \lambda_t)\mathbf{e}_t + \lambda_t\mathbf{S}. \tag{34}$$

Using $\mathbb{E}[\mathbf{S} \mid \mathcal{G}_t] = \mathbf{p}$ and $\mathrm{Var}(\mathbf{S} \mid \mathcal{G}_t) = \mathrm{diag}(\mathbf{p}) - \mathbf{p}\mathbf{p}^\top$, we obtain the *conditional* mean and variance:

$$\mathbb{E}[\mathbf{v}_{\mathrm{rvf}} \mid \mathcal{G}_t] = (1 - \lambda_t)\mathbf{e}_t + \lambda_t\mathbf{p} = \mathbf{v}_{\mathrm{vf}}, \tag{35}$$

$$\mathrm{Var}(\mathbf{v}_{\mathrm{rvf}} \mid \mathcal{G}_t) = \lambda_t^2\big(\mathrm{diag}(\mathbf{p}) - \mathbf{p}\mathbf{p}^\top\big). \tag{36}$$

#### A.5.4 COMPARISON VIA THE LAW OF TOTAL VARIANCE

Applying $\mathrm{Var}(\mathbf{X}) = \mathbb{E}[\mathrm{Var}(\mathbf{X} \mid \mathcal{G}_t)] + \mathrm{Var}(\mathbb{E}[\mathbf{X} \mid \mathcal{G}_t])$ to both updates yields

$$\mathrm{Var}(\mathbf{v}_{\mathrm{rvf}}) = \mathbb{E}[\mathrm{Var}(\mathbf{v}_{\mathrm{rvf}} \mid \mathcal{G}_t)] + \mathrm{Var}(\mathbb{E}[\mathbf{v}_{\mathrm{rvf}} \mid \mathcal{G}_t]) \tag{37}$$

$$= \mathbb{E}\big[\lambda_t^2\big(\mathrm{diag}(\mathbf{p}) - \mathbf{p}\mathbf{p}^\top\big)\big] + \mathrm{Var}(\mathbf{v}_{\mathrm{vf}}). \tag{38}$$

Hence,

$$\mathrm{Var}(\mathbf{v}_{\mathrm{rvf}}) - \mathrm{Var}(\mathbf{v}_{\mathrm{vf}}) = \mathbb{E}\big[\lambda_t^2\big(\mathrm{diag}(\mathbf{p}) - \mathbf{p}\mathbf{p}^\top\big)\big] \succeq \mathbf{0}, \tag{39}$$

because $\mathrm{diag}(\mathbf{p}) - \mathbf{p}\mathbf{p}^\top$ is positive semidefinite and expectations preserve the PSD order. The inequality is strict whenever $\mathbb{P}(\lambda_t \neq 0, \ \mathbf{p} \text{ is not one-hot}) > 0$.

**Coordinate-wise Form**

$$\mathrm{Var}\Big(\mathbf{v}_{\mathrm{rvf}}^{(i)}\Big) - \mathrm{Var}\Big(\mathbf{v}_{\mathrm{vf}}^{(i)}\Big) = \mathbb{E}\big[\lambda_t^2\, p_i(1 - p_i)\big] \geq 0, \tag{40}$$

with strict inequality under the same nondegeneracy conditions.

**Takeaway** Conditionally on $\mathcal{G}_t$, vf and rvf share the same mean, but rvf adds the covariance $\lambda_t^2(\operatorname{diag}(\mathbf{p}) - \mathbf{p}\mathbf{p}^\top)$. Unconditionally, rvf inherits the same across-trajectory variability as vf and *adds* a PSD term, so $\operatorname{Var}(\mathbf{v}_{\mathrm{rvf}}) \succeq \operatorname{Var}(\mathbf{v}_{\mathrm{vf}})$.

### A.6 NUMERICAL STABILITY ANALYSIS

In this section,we demonstrate that the vf-denoiser amplifies the model prediction error $\delta$ through the scaling factor $\lambda_t$, whereas the rvf-denoiser's sampling mechanism decouples this interaction, strictly bounding the numerical error to machine precision $\epsilon$.

#### A.6.1 PRELIMINARIES: ERROR DECOMPOSITION AND PROJECTION LEMMA

**Decomposition of Error Sources.** Let $\hat{p} = p + \delta$ be the neural network output, where $\delta$ denotes the statistical error of the model.

We define three update vectors: the **true vector** $\mathbf{v}_{\mathrm{true}}$ derived from $p$; the **statistical vector** $\mathbf{v}_{\mathrm{stat}}$ derived from $\hat{p}$ before projection; and the **numerical vector** $\mathbf{v}_{\mathrm{num}}$, the actual output after projection $\Pi$ and floating-point arithmetic. The total error can be decomposed into a statistical and a numerical part:

$$\left\|\mathbf{v}_{\mathrm{num}} - \mathbf{v}_{\mathrm{true}}\right\|_1 \leq \underbrace{\left\|\mathbf{v}_{\mathrm{stat}} - \mathbf{v}_{\mathrm{true}}\right\|_1}_{\text{Statistical Error}} + \underbrace{\left\|\mathbf{v}_{\mathrm{num}} - \mathbf{v}_{\mathrm{stat}}\right\|_1}_{\text{Numerical Error}}. \tag{41}$$

**Since the vf-denoiser and rvf-denoiser share the same underlying transition kernel, their statistical error components are strictly identical.** In this section, **we focus exclusively on the numerical error**, isolating the deviation introduced solely by the solver's execution mechanism.

**Projection Operator and Truncation Lemma.** Define the projection operator $\Pi : \mathbb{R}^K \to \Delta^{K-1}$ as "clipping negative entries and renormalizing":

$$\Pi(u) := \frac{\max(0, u)}{\sum_i \max(0, u_i)}, \tag{42}$$

where $\max(0, u)$ is applied elementwise.

Define the *truncation mass* $L(u)$ of a vector $u$ as the sum of the absolute values of all negative components:

$$L(u) := \sum_{u_i < 0} |u_i|. \tag{43}$$

**Lemma 1 (Projection Error Identity).** For any vector $u$ with $\sum_i u_i = 1$, if $\Pi(u)$ is well-defined, then the $L_1$-error introduced by the projection operator equals twice the truncation mass:

$$\left\|\Pi(u) - u\right\|_1 = 2L(u). \tag{44}$$

*Proof.* The projection error consists of two components: the truncation of negative values, contributing $\sum_{u_i < 0} |u_i| = L(u)$; and the renormalization of nonnegative values (which sum to $1 + L(u)$). The latter contributes $(1 + L(u)) \left| \frac{1}{1+L(u)} - 1 \right| = L(u)$. Summing both yields a total error of $2L(u)$. $\square$

#### A.6.2 NUMERICAL INSTABILITY OF THE VF-DENOISER: LINEAR AMPLIFICATION OF NOISE

We first rewrite the vf update in terms of its conditional transition kernel. The ideal vf transition kernel at time $t$ with true posterior $p$ is

$$p_{t+h}^v(x \mid x_t) := (1 - \lambda_t)\,\mathbf{1}_{\{x=x_t\}} + \lambda_t\,p(x), \tag{45}$$

so that $p_{t+h}^v(\cdot \mid x_t)$ is a nonnegative probability vector on the simplex.

Given the approximate posterior $\hat{p} = p + \delta$, the ideal unprojected statistical vector of the vf-denoiser can be written as

$$\mathbf{v}_{\mathrm{vf}}^{\mathrm{stat}} = p_{t+h}^v(\cdot \mid x_t) + \lambda_t\,\delta. \tag{46}$$

In practice, floating-point arithmetic introduces a perturbation $\xi$ with $\|\xi\|_1 \leq \epsilon$. The actual vector input to the projection operator is

$$\tilde{\mathbf{v}}_{\mathrm{vf}} = \mathbf{v}_{\mathrm{vf}}^{\mathrm{stat}} + \xi = \underbrace{p_{t+h}^v(\cdot \mid x_t)}_{\geq 0} + (\lambda_t\,\delta + \xi). \tag{47}$$

The final numerical vector is $\hat{\mathbf{v}}_{\mathrm{vf}} = \Pi(\tilde{\mathbf{v}}_{\mathrm{vf}})$.

We define the numerical error as the deviation of the final output from the intended statistical vector $\mathbf{v}_{\mathrm{vf}}^{\mathrm{stat}}$. Using the triangle inequality, we decompose the error $\mathcal{E}_{\mathrm{vf}}^{\mathrm{num}}$:

$$\mathcal{E}_{\mathrm{vf}}^{\mathrm{num}} := \left\|\hat{\mathbf{v}}_{\mathrm{vf}} - \mathbf{v}_{\mathrm{vf}}^{\mathrm{stat}}\right\|_1 \tag{48}$$

$$\leq \underbrace{\left\|\Pi(\tilde{\mathbf{v}}_{\mathrm{vf}}) - \tilde{\mathbf{v}}_{\mathrm{vf}}\right\|_1}_{\text{Projection Error}} + \underbrace{\left\|\tilde{\mathbf{v}}_{\mathrm{vf}} - \mathbf{v}_{\mathrm{vf}}^{\mathrm{stat}}\right\|_1}_{\text{Floating-point Error}}. \tag{49}$$

By Lemma 1, the projection error equals $2L(\tilde{\mathbf{v}}_{\mathrm{vf}})$. The second term is simply $\|\xi\|_1$. Thus:

$$\mathcal{E}_{\mathrm{vf}}^{\mathrm{num}} \leq 2L(\tilde{\mathbf{v}}_{\mathrm{vf}}) + \|\xi\|_1. \tag{50}$$

To bound the truncation mass $L(\tilde{\mathbf{v}}_{\mathrm{vf}})$, note that the bracketed kernel $p_{t+h}^v(\cdot \mid x_t)$ is theoretically nonnegative. Hence any negative entries in $\tilde{\mathbf{v}}_{\mathrm{vf}}$ must originate from the noise term $\lambda_t\delta + \xi$. Using the property that $L(u + v) \leq L(u) + \|v\|_1$ (and $L(p_{t+h}^v(\cdot \mid x_t)) = 0$), we obtain

$$L(\tilde{\mathbf{v}}_{\mathrm{vf}}) \leq \left\|\text{negative part of } (\lambda_t\delta + \xi)\right\|_1 \leq \frac{1}{2}\lambda_t\|\delta\|_1 + \|\xi\|_1. \tag{51}$$

Substituting this back into the error bound:

$$\mathcal{E}_{\mathrm{vf}}^{\mathrm{num}} \leq 2\left(\frac{1}{2}\lambda_t\|\delta\|_1 + \|\xi\|_1\right) + \|\xi\|_1 = \lambda_t\|\delta\|_1 + 3\|\xi\|_1. \tag{52}$$

Letting $\eta$ be the upper bound of $\|\delta\|_1$ and $\epsilon$ be the machine precision bound on $\|\xi\|_1$, we obtain:

$$\mathcal{E}_{\mathrm{vf}}^{\mathrm{num}} \leq \lambda_t\eta + 3\epsilon. \tag{53}$$

*Conclusion.* Since the weighted statistical error typically dominates machine precision ($\lambda_t\eta \gg \epsilon$) in practical scenarios, the numerical error bound is effectively determined by the model error:

$$\mathcal{E}_{\mathrm{vf}}^{\mathrm{num}} \leq \mathcal{O}(\lambda_t\eta). \tag{54}$$

This indicates that the vf-denoiser directly amplifies the statistical prediction error, converting it into significant numerical bias.

### A.6.3 NUMERICAL ROBUSTNESS OF THE RVF-DENOISER: DECOUPLING VIA SPARSITY

We now express the rvf update in terms of its conditional transition kernel. Given a sampled target $z \sim \hat{p}$, the rvf-denoiser defines the *conditional* transition kernel

$$p_{t+h}^r(x \mid x_t, z) := (1 - \lambda_t)\,\mathbf{1}_{\{x=x_t\}} + \lambda_t\,\mathbf{1}_{\{x=z\}}, \tag{55}$$

which is a valid probability distribution. The corresponding ideal sparse update vector for this sample is therefore

$$\mathbf{v}_{\mathrm{rvf}}^{\mathrm{stat}}(z) = p_{t+h}^r(\cdot \mid x_t, z). \tag{56}$$

In practice, floating-point errors introduce a perturbation $\xi$ with $\|\xi\|_1 \leq \epsilon$, so the actual vector before projection is

$$\tilde{\mathbf{v}}_{\mathrm{rvf}} = \mathbf{v}_{\mathrm{rvf}}^{\mathrm{stat}}(z) + \xi = \underbrace{p_{t+h}^r(\cdot \mid x_t, z)}_{\geq 0} + \xi. \tag{57}$$

The final numerical vector is

$$\hat{\mathbf{v}}_{\mathrm{rvf}} = \Pi(\tilde{\mathbf{v}}_{\mathrm{rvf}}). \tag{58}$$

Using Lemma 1, the numerical error for a given sample $z$ is

$$\mathcal{E}_{\mathrm{rvf}}^{\mathrm{num}}(z) \coloneqq \left\| \hat{\mathbf{v}}_{\mathrm{rvf}} - \mathbf{v}_{\mathrm{rvf}}^{\mathrm{stat}}(z) \right\|_1 \leq 2\, L(\tilde{\mathbf{v}}_{\mathrm{rvf}}) + \|\xi\|_1. \tag{59}$$

Because $p_{t+h}^r(\cdot \mid x_t, z)$ is itself nonnegative, any negative components of $\tilde{\mathbf{v}}_{\mathrm{rvf}}$ must come from the floating-point error $\xi$. Hence

$$L(\tilde{\mathbf{v}}_{\mathrm{rvf}}) \leq \left\| \text{negative part of } \xi \right\|_1 \leq \|\xi\|_1 \leq \epsilon. \tag{60}$$

Therefore, for each $z$ we have

$$\mathcal{E}_{\mathrm{rvf}}^{\mathrm{num}}(z) \leq 2\epsilon + \epsilon = 3\epsilon, \tag{61}$$

and thus, up to a constant factor, the rvf numerical error is of order

$$\mathcal{E}_{\mathrm{rvf}}^{\mathrm{num}} \leq \mathcal{O}(\epsilon). \tag{62}$$

*Conclusion.* The numerical error of the rvf-denoiser is controlled solely by machine precision and is independent of the model error $\eta$. The "sample–then–sparsify" mechanism effectively decouples numerical error from the statistical prediction error.

### A.6.4 SUMMARY AND COMPARISON

Comparing the upper bounds of numerical error for the two denoisers, we obtain

$$\frac{\text{Upper bound of } \mathcal{E}_{\mathrm{vf}}^{\mathrm{num}}}{\text{Upper bound of } \mathcal{E}_{\mathrm{rvf}}^{\mathrm{num}}} \sim \frac{\lambda_t\, \eta}{\epsilon}. \tag{63}$$

In our experiments, the validation error $\eta$ of the model is typically around $10^{-2}$, whereas single-precision machine error $\epsilon$ is much smaller (often below $10^{-6}$).

Crucially, as $\lambda_t$ grows, the vf-denoiser directly amplifies the prediction error $\eta$, resulting in numerical noise far exceeding machine precision. In contrast, the rvf-denoiser structurally decouples this interaction, keeping the error strictly bound by $\epsilon$ and ensuring superior numerical robustness.

## B EXPERIMENTAL DETAILS

### B.1 COMPUTING ENVIRONMENT

Our implementation is based on PyG (Fey & Lenssen, 2019). The experiments are conducted on a single workstation with 8 A100 GPUs.

### B.2 COMPUTATIONAL COST ANALYSIS

In this section, we address the concern regarding the potential computational overhead introduced by the rvf-denoiser. Although the rvf-denoiser involves an additional sampling step compared to the vf-denoiser, we demonstrate both empirically and theoretically that this cost is negligible.

### B.2.1 EMPIRICAL RUNTIME COMPARISON

We conducted a rigorous runtime comparison on four datasets: Planar, Tree, SBM, and QM9. As shown in Table 7, the wall-clock time differences between vf and rvf are statistically insignificant. In some cases (e.g., QM9), rvf appears slightly faster solely due to system-level fluctuations (such as GPU scheduling jitter and memory allocation noise), which overshadow the minute computational difference between the two methods.

### B.2.2 THEORETICAL COMPLEXITY ANALYSIS

To further justify the minimal overhead, we provide a time complexity analysis. Let $N$ be the number of nodes, $L$ the number of Transformer layers, and $d$ the hidden dimension.

- **Model Inference ($T_{\mathbf{model}}$):** The computational bottleneck lies in the self-attention mechanism of the graph transformer, which scales as:

$$T_{\mathrm{model}} \approx \mathcal{O}(L \cdot N^2 \cdot d). \tag{64}$$

Table 7: Runtime comparison between vf and rvf samplers. The results indicate no observable latency overhead for the rvf-denoiser.

| Dataset | Graphs Sampled | vf-denoiser Sampling Time (s) | rvf-denoiser Sampling Time (s) |
|---------|----------------|-------------------------------|--------------------------------|
| Planar | 40 | $24.8 \pm 0.1$ | $24.8 \pm 0.1$ |
| Tree | 40 | $2.6 \pm 0.1$ | $2.6 \pm 0.1$ |
| SBM | 40 | $135.5 \pm 8.3$ | $135.9 \pm 8.2$ |
| QM9 | 10,000 | $126.5 \pm 0.2$ | $125.7 \pm 0.2$ |

- **Sampling Overhead ($T_{\textbf{sample}}$):** Sampling a discrete adjacency matrix involves iterating over $N^2$ edges. Both vf and rvf require $O(N^2)$ operations to compute the update. rvf performs one additional sampling step from the categorical distribution, adding another $O(N^2)$ term. The relative overhead ratio is:

$$\frac{\text{Extra Cost}}{T_{\text{model}}} \approx \frac{\mathcal{O}(N^2)}{\mathcal{O}(L \cdot N^2 \cdot d)} = \frac{1}{L \cdot d}. \tag{65}$$

**Conclusion:** Under typical experimental settings (e.g., $d = 256$, $L = 10$), the theoretical additional cost is less than $0.04\%$. This confirms that the rvf-denoiser improves generation diversity without incurring any practical computational penalty.

### B.3 IMPLEMENTATION DETAILS

We adopt the Graph Transformer backbone from DiGress (Vignac et al., 2022), with further experimental details available in our source code at https://anonymous.4open.science/r/SimGFM-F9C5.

### B.3.1 SPECIFICATION OF SOURCE DISTRIBUTION $p_0$

To ensure full reproducibility, we explicitly specify the source distribution $p_0$ used for initialization in each experiment. The choice of $p_0$ defines the prior noise distribution from which the backward generation process starts ($x_1 \sim p_0$).

Table 8: Source distribution ($p_0$) configurations for all datasets.

| Dataset | Node Distribution ($p_0^V$) | Edge Distribution ($p_0^E$) | Remarks |
|---------|------------------------------|------------------------------|---------|
| QM9 | Marginal | Marginal | — |
| QM9H | Marginal | Marginal | — |
| Planar | Marginal | Marginal | — |
| Tree | Marginal | Marginal | — |
| MOSES | Marginal | Marginal | — |
| Ego-Small | Marginal | Marginal | — |
| Community-Small | Marginal | Marginal | — |
| Grid | Marginal | Marginal | — |
| TLS | Marginal | Marginal | — |
| SBM | AbsorbFirst | AbsorbFirst | Initialized with absorbing state |

## C FURTHER RESULTS

### C.1 SCHEDULER SENSITIVITY

We adopt the power-accelerated family $\kappa_t = f_k(t) = 1 - (1 - t)^k$ with $k \in \{1, 2, 5, 10, 20\}$, where larger $k$ front loads progress. Table 9 reports Valid under three representative settings: QM9 with a small step budget (10 steps), MOSES with a large step budget (200 steps), and TLS conditional generation (200 steps). On QM9 (10 steps), Valid improves monotonically with $k$ and peaks at

$k = 20$, suggesting strong front loading is preferred when steps are scarce. On MOSES (200 steps), Valid peaks at $k = 10$ and remains close at $k = 20$, indicating that moderate front loading balances early progress and late refinement. On TLS (200 steps), the best results occur at $k = 2$ and $k = 20$, while $k = 10$ underperforms, reflecting task-dependent optima.

Table 9: Scheduler sensitivity on QM9 (10 steps), MOSES (200 steps), and TLS (200 steps).

| Dataset | Steps | Valid ↑ | | | | |
|---------|-------|---------|---------|---------|---------|---------|
| | | $\kappa_t = t$ | $\kappa_t = f_2(t)$ | $\kappa_t = f_5(t)$ | $\kappa_t = f_{10}(t)$ | $\kappa_t = f_{20}(t)$ |
| QM9 | 10 | 95.72 | 95.84 | 98.83 | 99.09 | **99.37** |
| MOSES | 200 | 81.92 | 87.52 | 88.24 | **89.39** | 89.29 |
| TLS | 200 | 93.75 | **96.25** | 95.00 | 93.75 | **96.25** |

We further compare our scheduler to the identity baseline, which assumes uniform transition rates. Table 10 shows results for both *vf-denoiser* and *rvf-denoiser* across four datasets. Our scheduler consistently improves validity, underscoring the importance of allocating more updates to the refinement phase and validating the effectiveness of our rate-matrix design.

Table 10: Ablation study on the transition rate matrix. We compare the performance of the identity scheduler versus our proposed scheduler using both vf-denoiser and rvf-denoiser. The results (validity %) demonstrate the critical role of our rate matrix design.

| Method | QM9 | QM9H | Tree | Planar |
|--------|-----|------|------|--------|
| vf-denoiser (w/ identity scheduler) | $98.3 \pm 0.2$ | $97.7 \pm 0.1$ | $49.5 \pm 4.0$ | $38.0 \pm 5.0$ |
| rvf-denoiser (w/ identity scheduler) | $99.3 \pm 0.1$ | $95.6 \pm 0.1$ | $57.0 \pm 4.6$ | $57.5 \pm 6.5$ |
| vf-denoiser (w/ our scheduler) | $99.6 \pm 0.0$ | $97.7 \pm 0.1$ | $95.5 \pm 1.0$ | $96.0 \pm 3.0$ |
| rvf-denoiser (w/ our scheduler) | $99.8 \pm 0.0$ | $98.4 \pm 0.1$ | $99.5 \pm 1.0$ | $100.0 \pm 0.0$ |

# D  FURTHER DISCUSSION

## D.1  LIMITATIONS AND IMPACT

We have not fully explored the space of DFM schedulers, leaving room for improvement. As with all molecular generators, practitioners must ensure responsible downstream use; our focus is methodological efficiency, not property-targeted design.

## D.2  REPRODUCIBILITY STATEMENT

We have taken several steps to ensure reproducibility. Source code, datasets, and detailed instructions are available at https://anonymous.4open.science/r/SimGFM-F9C5.

## D.3  LLM USAGE

We used large language models (LLMs) for language editing and polishing only.

## D.4  ETHICS STATEMENT

Our study does not involve human subjects, sensitive personal data, or applications with foreseeable harmful impact. All datasets used are publicly available, and we follow community standards regarding data usage, fairness, and privacy.

