# OpenReview forum: "SimGFM: Simplifying Discrete Flow Matching for Graph Generation"
_ICLR.cc/2026/Conference — Submitted to ICLR 2026_

### Official Review · Reviewer_nKK3 · 2025-10-22

**Soundness:** 4
**Presentation:** 4
**Contribution:** 4
**Rating:** 8
**Confidence:** 4

**Summary:**

The paper introduces SimGFM (Simplified Graph Flow Matching), a novel approach for graph generation based on discrete flow matching (DFM). SimGFM's core innovation is a graph-structured rate formulation grounded in minimalist design principles, specifically aiming for a clear mathematical expression that is free of ad-hoc heuristics and fully consistent with the continuity equation. The model demonstrates a significant empirical advantage, achieving competitive performance on the QM9 dataset using only 50 sampling steps (a process that previously required 500−1000 steps) and matching or surpassing baseline models on most datasets at just 50 steps. This demonstrates a compelling blend of efficiency and competitive generative capability.

**Strengths:**

1. **Substantial Advancement in Discrete Graph Flow Matching Efficiency:** The proposed SimGFM model addresses a critical and long-standing challenge in discrete graph flow matching: the computational burden imposed by a large number of sampling steps. By introducing an elegant, simplified rate formulation, the model achieves a significant reduction in the required sampling steps while demonstrably maintaining competitive performance across diverse graph generation benchmarks.

2. **Exemplary Clarity and Presentation:** The manuscript is exceptionally well-structured and written. The authors effectively communicate complex theoretical concepts, the proposed methodology, and the empirical results.

**Weaknesses:**

1. **Perceived Simplicity of Core Methodology**: While the reviewer appreciates the elegance of the proposed method, some readers might find the technical contribution to be relatively straightforward, as the core innovations primarily revolve around a refinement of the rate-matrix estimator and the time scheduler.

**Questions:**

1. There is a potential issue with the citation of the CatFlow model in the manuscript. Could the authors please verify the accuracy and completeness of the reference corresponding to the CatFlow work?

I have no further questions for authors.

---

> ### Author Response · Authors · 2025-11-23
>
> **We sincerely appreciate your recognition of our contributions!**
>
> **(Response to W1): Summary of Contributions**
> > W1: "Perceived Simplicity of Core Methodology: While the reviewer appreciates the elegance of the proposed method, some readers might find the technical contribution to be relatively straightforward, as the core innovations primarily revolve around a refinement of the rate-matrix estimator and the time scheduler."
>
> Thank you very much for your thoughtful feedback and for recognizing the elegance of our proposed method. We appreciate your observation regarding the perceived straightforwardness of our technical contribution, as also noted by Reviewer onhW.
>
> We fully acknowledge this point. Our intention was not to introduce architectural complexity, but rather to design a method that remains conceptually clean and practically efficient. We believe this simplicity is a key strength of SimGFM, as it enables a substantial reduction in sampling steps—from the typical 500–1000 to around 10–50—while still maintaining competitive performance across benchmarks. We view this balance between simplicity and effectiveness as central to our contribution.
>
> In addition, we took these comments seriously and carefully examined whether our framework offers deeper insights beyond procedural refinement. As summarized in our response to reviewer onhW Q3, further analysis conducted during the rebuttal period revealed a meaningful observation about discrete flow dynamics: **under uniform denoising, valid graph structures predominantly emerge near the end of the denoising trajectory**.
>
> This pattern carries important implications.
>
> 1. It suggests that graph generative processes may benefit from sampling at output values of the scheduler function $f(t)$ that are close to 1.
> 2. It motivates using scheduler functions—such as high-order polydec schedules—that expose the model to values near 1 earlier in the trajectory, thereby accelerating the formation of valid graph structures.
>
> Our theoretical analysis further shows that high-order schedulers (e.g., $k = 20$) remain numerically stable for vf/rvf denoisers but deteriorate under Campbell's construction, whose updates shrink too quickly as $t \to 1$. This distinction provides a clear explanation for **why SimGFM can achieve high validity with substantially fewer denoising steps**.
>
> We believe these empirical and theoretical findings contribute new insight into the behavior of discrete flows in graph generation, and the revised paper now incorporates these results and analyses.
>
>
> **(Response to Q1): The Citation of CatFlow**
> > Q1: There is a potential issue with the citation of the CatFlow model in the manuscript. Could the authors please verify the accuracy and completeness of the reference corresponding to the CatFlow work?
>
> Thank you very much for bringing this to our attention. We sincerely apologize for the oversight.
>
> 1. We have carefully reviewed the CatFlow paper and confirmed that the metric reported for the QM9 dataset is indeed **Valid**, not **Relaxed Valid**. We have corrected this in the revised manuscript.
>
> 2. Additionally, we have included the results of CatFlow on the Ego-small, Community-small, and Grid datasets in Table 3 of the revised manuscript to ensure a more comprehensive evaluation.

---

> > ### Comment · Reviewer_nKK3 · 2025-11-27
> >
> > Thanks for your detailed reply. I will maintain my initial score, and good luck to you

---

> > > ### Author Response · Authors · 2025-11-28
> > >
> > > Dear Reviewer nKK3,
> > >
> > > Thank you for taking the time to review our rebuttal. We sincerely appreciate your positive and insightful assessment, as well as your reaffirmation of the rating. We are also grateful for your encouraging response.
> > >
> > > Best regards,
> > >
> > > The Authors

---

### Official Review · Reviewer_UM94 · 2025-10-23

**Soundness:** 1
**Presentation:** 3
**Contribution:** 1
**Rating:** 2
**Confidence:** 3

**Summary:**

This paper introduces SimGFM, a discrete flow matching generative model for graph generation. The work distinguishes between two concurrent formulations of Discrete Flow Matching (DFM): the formulation of Campbell et al. and that of Gat et al. (also referred to as the VF-denoiser). Unlike previous DFM paper based on Campbell’s formulation, this paper builds upon the Gat et al. variant, which is presented as conceptually simpler and more flexible.

The authors identify a potential limitation in the deterministic updates of the VF-denoiser, which they claim can lead to mode collapse and reduced sample diversity. To address this, the paper introduces a random VF-denoiser (rvf-denoiser), which replaces the claimed deterministic update by sampling from the denoiser’s probability distribution and before updating the graph using a Dirac delta with mass at the sampled graph. This modification is intended to promote exploration and improve diversity during generation.

Extensive experiments on multiple datasets are presented, with the proposed method reported to improve state-of-the-art results on most metrics.

**Strengths:**

The paper is clearly written and easy to follow.

The mathematical propositions and proofs are well-structured, intuitive, and appear to be correct.

The experimental section is extensive, covering a range of datasets and showing strong empirical performance across several benchmarks.

**Weaknesses:**

### Main Claims
The paper’s primary contribution is the introduction of the random VF-denoiser (rvf-denoiser) in place of the standard vf-denoiser. However, it is not clear that the two procedures are different. Based on the formulation, the rvf-denoiser appears to be mathematically equivalent to the VF-denoiser:

   $p_{s | t}^{\text{rvf}}(x^i | x_t) = E_{x^i_{1|t} \sim p_\theta(x_1 | x_t)}[p(x_s | x^i_{1|t}, x_t)]
    = E_{x^i_{1|t}}[\delta_{x_t}(x_s) + h u^{rvf}(x^i, x_t)]
    = \delta_{x_t}(x_s) + hE_{x^i_{1|t}}[u^{rvf}(x^i, x_t)]
    = \delta_{x_t}(x_t) + hu(x^i, x_t)
    = p^{vf}_{s | t}(x^i | x_t)$

If this reasoning holds, the rvf-denoising step is equivalent to the vf-denoising step, raising doubts about the novelty and motivation of the approach.

The paper presents the VF-denoiser as deterministic (lines 56 and 207), yet in practice, $G_{t+\Delta_t}$ is randomly sampled at each denoising step. This apparent inconsistency should be clarified.

The authors argue that the rvf-denoiser “allows the trajectory to commit early to one alternative and break symmetry,” thereby enhancing exploration and diversity (lines 233–238). Such claims should be supported by either a theoretical argument or dedicated empirical evidence demonstrating that rvf-denoising indeed leads to improved diversity without sacrificing sample quality.

Finally, the paper states that rvf and vf have similar computational costs. However, since the rvf-denoiser requires two samplings per denoising step, the additional cost may not be negligible. Providing quantitative runtime comparisons would substantiate this claim.

### Related Work
The recent paper [1] appears closely related, as it also employs an iterative denoising framework involving sampling $x_{1|x}$ before reconstruction. A more detailed discussion of how SimGFM differs from this approach would help clarify the novelty and positioning of the proposed method, as well as potentially clarify the distinction between VF and rVF.

### Empirical Evaluation
If vf and rvf are indeed distinct methods (and not equivalent formulations), the experimental section should include a systematic comparison between them to validate the claimed benefits of the rvf-denoiser.

For reproducibility, the paper should indicates the source distribution $p_0$ used for each experiment (for nodes and edges).

In addition, the results reported for the Ego-Small dataset appear to outperform the training set itself, which should not be possible. This discrepancy likely indicates a presentation or evaluation issue and should be clarified.

For the QM9 datasets (both with and without hydrogen), the paper does not specify which version is used (kekulized vs. aromatic bonds). This distinction is crucial, as results across these versions are not directly comparable.

------
[1] Y. Boget. Simple and critical iterative denoising: A recasting of discrete diffusion in graph gen-
eration. In Proceedings of the 42th International Conference on Machine Learning, Proceedings
of Machine Learning Research. PMLR, July 2025.

**Questions:**

Are rvf and vf really different or two equivalent formulartions of the same model?

Why the authors state that the vf-denoiser is deterministic?

How the authors explain that the results for ego-small outperform the training set?

Out of curiosity, what does vf stands for?

---

> ### Author Response · Authors · 2025-11-23
>
> Thank you for your constructive feedback. We appreciate the opportunity to **clarify any potential sources of confusion or misunderstanding**. We hope that the following explanations will provide greater clarity and prompt a reassessment of our work.
>
> **(Response to Q4): The meaning of vf**
> >Q4: Out of curiosity, what does vf stands for?
>
> "vf" stands for velocity field. In the context of our paper, when "vf" appears on its own, it specifically refers to the vf-denoiser model, and similarly, "rvf" refers to the rvf-denoiser.
>
> **(Response to W1 & W2 & W4 & W7 &Q1): Why rvf & vf are Different And Why rvf Enhancing Exploration and Diversity**
> > W1: The paper's primary contribution is the introduction of the rvf-denoiser in place of the standard vf-denoiser. However, it is not clear that the two procedures are different. Based on the formulation, the rvf-denoiser appears to be mathematically equivalent to the vf-denoiser.
> >
> > W2: If this reasoning holds, the rvf-denoising step is equivalent to the vf-denoising step, raising doubts about the novelty and motivation of the approach.
> >
> > W4: The authors argue that the rvf-denoiser "allows the trajectory to commit early to one alternative and break symmetry," thereby enhancing exploration and diversity. Such claims should be supported by either a theoretical argument or dedicated empirical evidence.
> >
> > W7: If vf and rvf are indeed distinct methods (and not equivalent formulations), the experimental section should include a systematic comparison between them to validate the claimed benefits of the rvf-denoiser.
> >
> > Q1: Are rvf and vf really different or two equivalent formulartions of the same model?
>
> Thank you for raising these important questions. The concerns focus on whether rvf and vf are truly different, whether rvf provides meaningful benefits. Below, we address these points in detail.
>
>
> **1. Variance Distinguishes rvf from vf**
>
> First, rvf and vf are valid DFM solvers, but operate through fundamentally different execution mechanisms. While they share the same expectation, they are distinct in terms of **variance**:
>
> $
> \operatorname{Var}\big[\delta_{x_t}(\cdot)+h\,u^{rvf}(\cdot,x_t)\big]
> \neq \operatorname{Var}\big[\delta_{x_t}(\cdot)+h\,u^{vf}(\cdot,x_t)\big].
> $
>
> This inequality holds because $u^{rvf}$ incorporates stochastic sampling, resulting in higher randomness compared to the deterministic expectation used in $u^{vf}$. We formalize this in the main paper as:
>
> **Proposition 2 (Variance Characterization of rvf-denoiser).**  Conditioned on the current state $x_t$, the one-step update of the rvf-denoiser exhibits strictly higher variance than that of the vf-denoiser in the sense of positive semi-definite (PSD) matrices. Specifically:
>
> $
> \operatorname{Var}(\delta_{x_t} + h u^{rvf}) \succeq \operatorname{Var}(\delta_{x_t} + h u^{vf})
> $
>
> The full proof is provided in Appendix A.5. This variance difference reflects distinct execution processes, leading to the performance differences in our ablation studies:
>
> |Valid↑|Step|QM9|QM9H|Planar|SBM|Tree|
> |-|-|-|-|-|-|-|
> |rvf|200|99.8$\pm$0.0|98.4$\pm$0.1|100.0$\pm$0.0|90.5$\pm$4.0|99.5$\pm$1.0|
> |vf |200|99.6$\pm$0.0|97.7$\pm$0.1|96.0$\pm$3.0|88.5$\pm$3.0|95.5$\pm$1.0|
>
> Given these consistent empirical improvements, a plausible explanation is that the stochasticity inherent in this variance relates to the mitigation of compounding denoising errors(detailed in response to W6).
>
> **2. Expectation Equivalence Implies Consistency, Not Identity**
>
> The reviewer correctly notes that the expected values are identical. According to the derivation by Itai et al., any valid DFM velocity matrix must satisfy the **continuity equation** in expectation [2]:
>
> $ \begin{array}{l} \mathbb{E}\_{X\_{t}} \prod\_{i=1}^{N} {\left[\delta_{X\_{t}}\left(x^{i}\right)+h u\_{t}^{i}\left(x^{i}, X\_{t}\right)\right]=\mathbb{E}\_{X\_{t}}\left[\delta\_{X\_{t}}(x)+h \sum\_{i=1}^{N} \delta\_{X\_{t}}\left(x^{\bar{i}}\right) u\_{t}^{i}\left(x^{i}, X\_{t}\right)\right]+o(h) } \\\\ = p\_{t}(x)-h \operatorname{div}\_{x}\left(p\_{t} u\_{t}\right)+o(h) {=} p_{t}(x)+h \dot{p}\_{t}(x)+o(h)=p\_{t+h}(x)+o(h), \end{array} $
>
> Satisfying this equation is a *necessary condition* for validity, meaning there are infinite valid rate matrices. The fact that rvf shares the same expectation as vf confirms that **rvf is a valid DFM solver**, but it does not imply that the two algorithms are identical in practice.
>
> Regarding the formula you proposed, the correct derivation is as follows:
>
> $
> E_{x_t}\\!\\left[\\delta_{x_t}(\\cdot)+h\\,u^{rvf}(\cdot,x_t)\\right]
> = \\delta_{x_t}(\\cdot)+h\\,E_{x_t}\\!\\left[u^{rvf}(\\cdot,x_t)\right]
> = \\delta_{x_t}(\\cdot)+h\\,E_{x_t}\\!\\left[u^{vf}(\\cdot,x_t)\right]
> = E_{x_t}\\!\\left[\\delta_{x_t}(\\cdot)+h\\,u^{vf}(\\cdot,x_t)\right]
> $
>
> However, this equality in expectation simply proves the mathematical legitimacy of rvf, while the difference in variance alters the actual trajectories, directly driving the performance improvement.

---

> > ### Comment · Reviewer_UM94 · 2025-11-26
> >
> > I would like to thank the authors for their responses. Unfortunately, I still have a major concern regarding the principal claim of the paper. In particular, I continue to believe that VF and RVF are strictly equivalent. I therefore kindly ask the authors to indicate and explain, in the derivation below, which equality they consider invalid. For clarity, I have numbered the equations. I denote by $p^R_{t+h}$ the RVF update distribution and by  $p^V_{t+h}$  the VF update distribution
> >
> > 1. $p^R_{t+h}(x \mid x_t) = \delta_{x_t}(x)+\lambda_t(\delta_{x_{1|t}}(x) - \delta_{x_t}(x))$, with $x_{1|t} \sim p_{1|t}(x|x_t)$ , by the definition of an RVF denoising step given in Eq. 11 and Algorithm 1 (with $\lambda_t$ as defined in A.5.1).
> >
> > 2.  $p^R_{t+h}(x \mid x_t) = E_{x_{1|t}} \left[\delta_{x_t}(x)+hu^{rvf}(x, x_t)\right]$. This rewrites (1) as a single expression by setting $hu^{rvf}(\cdot,x_t)= \lambda_t(\delta_{x_{1|t}}(x) - \delta_{x_t}(x))$ and expressing sampling as an expectation.
> >
> > 3. $E_{x_{1|t}} \left[\delta_{x_t}(x)+hu^{rvf}(x, x_t)\right] =  \delta_{x_t}(x)+hE_{x_{1|t}} \left[u^{rvf}(x,x_t)\right]$ by taking the constant term outside the expectation.
> >
> > 4. $\delta_{x_t}(x)+hE_{x_{1|t}} \left[u^{rvf}(x,x_t)\right] =  \delta_{x_t}(x)+h u^{vf}(x,x_t)$, by Proposition 1 (Eq. 12).
> >
> > 5. $ \delta_{x_t}(x)+h u^{vf}(x,x_t) = p^V_{t+h}(x \mid x_t)$ by definition (Eq. 2)
> >
> > Putting everything together, we obtain $p^R_{t+h} = p^V_{t+h}$ and therefore RVF is strictly equivalent to VF.
> >
> > I am genuinely interested in understanding whether this reasoning is incorrect, and I would appreciate a detailed explanation from the authors indicating precisely where the derivation fails.

---

> ### Author Response · Authors · 2025-11-23
>
> **(Response to W6): Compare with SID/CID**
> > W6: A more detailed discussion of how SimGFM differs from [1] would help clarify the novelty and positioning of the proposed method, as well as potentially clarify the distinction between VF and rVF.
>
> Thank you for highlighting this important reference. We have included a detailed comparison and discussion with **SID/CID [1]** in the revised manuscript.
>
> As discussed in the referenced paper, **"Compounding Denoising Errors"** is a common challenge faced by iterative generative models such as Diffusion and Flow Matching. From this perspective, SID/CID and SimGFM aim for the same goal via different but effective pathways:
>
> 1.  **SID/CID (Architectural Perspective):** SID introduces a novel and concise architecture that predicts $x_1$ from $x_t$ and re-noises it to obtain $x_{t+h}$, thereby mitigating drift during iteration. Additionally, CID incorporates a Critic to further correct model predictions, significantly boosting performance.
> 2.  **SimGFM (Dynamics Perspective):** Our method mitigates error accumulation from a dynamics standpoint through **scheduler design** and **rvf sampling**:
>     *   **Scheduler:** Our selected Scheduler increases the scaling factor $\frac{\dot{\kappa_t}}{1-\kappa_t} \cdot h$ during critical generation stages, thereby numerically reducing error propagation.
>     *   **Rvf-denosier:** More critically, rvf's two-stage sampling mechanism prevents the scale factor from directly amplifying the model prediction error. This structural difference effectively curbs error accumulation, as formally derived in Appendix A.6 in the revised paper.
>
> While SID avoids dependence on intermediate states through architectural design, SimGFM leverages both an increased scaling factor and rvf’s two-stage sampling mechanism to reshape denoising trajectories and mitigate compounding denoising errors.
>
> As shown in the table below, SimGFM achieves highly competitive performance on the QM9 dataset compared to SID/CID.
>
> |Method|QM9|QM9|
> |-|-|-|
> ||Step↓|Valid↑|
> |SID|-|99.7|
> |CID|-|99.9|
> |vf(w/o scheduler)|200|98.3$\pm$0.2|
> |rvf(w/o scheduler)|200|99.3$\pm$0.1|
> |SimGFM(vf)|200|99.6$\pm$0.0|
> |SimGFM(rvf)|200|99.8$\pm$0.0|
>
> **This conclusion is strongly supported by empirical evidence:** As shown in the table above, in the absence of our optimized Scheduler ($\kappa_t=1-(1-t)^k,k=20$), the performance of vf collapses from 99.6% to 98.3%, whereas rvf maintains a high validity of 99.3%. This demonstrates the inherent robustness of rvf against error accumulation. In essence, the improvement of rvf over vf parallels the advancement of CID over SID: both aim to achieve more robust generative trajectories within the discrete space.
>
> **(Response to W5): Computational Cost**
> > W5:"Finally, the paper states that rvf and vf have similar computational costs. However, since the rvf-denoiser requires two samplings per denoising step, the additional cost may not be negligible. Providing quantitative runtime comparisons would substantiate this claim."
>
> Rvf differs from vf by only a single additional line of implementation, and the extra computational cost introduced is extremely small. Empirical evidence shows that the difference in sampling time between the two methods is practically unobservable (and on QM9, rvf even appears slightly faster than vf), far below the natural runtime fluctuations inherent to the sampling process.
>
> | Dataset | Graphs Sampled | vf Sampling Time(s) | rvf Sampling Time(s) |
> |-|-|-|-|
> |Planar|40|24.8$\pm$0.1|24.8$\pm$0.1|
> |Tree|40|2.6$\pm$0.1|2.6$\pm$0.1|
> |SBM|40|135.5$\pm$8.3|135.9$\pm$8.2|
> |QM9|10000|126.5$\pm$0.2|125.7$\pm$0.2|
>
> From a theoretical standpoint, rvf adds only a lightweight numerical operation relative to vf, whose cost is negligible compared with the dominant components of runtime—such as model forward computation, tensor operations, memory scheduling, and stochastic sampling. In practice, wall-clock time is also affected by system-level variability (e.g., GPU scheduling jitter, memory allocation, and asynchronous execution), which easily exceeds the tiny overhead introduced by rvf.
>
> For completeness, we provide a brief time-complexity comparison.
>
> Let $N$ be the number of nodes, $L$ the number of Transformer layers, and $d$ the hidden dimension.
>
> 1.  Model Inference ($T_{\text{model}}$): The computational cost is dominated by the Self-Attention mechanism, which scales as $O(L \cdot N^2 \cdot d)$.
> 2.  Sampling Overhead ($T_{\text{sample}}$): Sampling a discrete adjacency matrix involves iterating over $N^2$ edges, scaling as $O(N^2)$.
> 3.  Comparison: rvf introduces only one additional sampling step compared to vf. The relative overhead is:
>
>     $ \frac{\text{Extra Cost}}{\text{Model Inference}} \approx \frac{O(N^2)}{O(L \cdot N^2 \cdot d)} = \frac{1}{L \cdot d} $
>
> With typical settings (e.g., $d=256, L=10$), the additional cost is less than 0.04%, rendering it mathematically and practically negligible.

---

> > ### Author Response · Authors · 2025-11-23
> >
> > **(Response to W3): The VF-denoiser as deterministic**
> > > W3:The paper presents the VF-denoiser as deterministic (lines 56 and 207), yet in practice, $G_{t+\Delta t}$ is randomly sampled at each denoising step. This apparent inconsistency should be clarified.
> >
> > Thank you for pointing this out. We acknowledge that our initial phrasing may have been imprecise. Our intention was to convey that rvf exhibits significantly higher stochasticity compared to vf due to its larger variance. We have revised this description in the manuscript.
> >
> > **(Response to  W8): Distribution $p_0$**
> > > W8: For reproducibility, the paper should indicates the source distribution $p_0$ used for each experiment (for nodes and edges).
> >
> > The specific selection of $p_{0}$ is included in the code provided in the anonymous repository. Furthermore, we have added a section to the Appendix to explicitly describe the choice of $p_{0}$.
> >
> >
> > **(Response to Empirical W9 & Q3): Ego-Small Performance**
> > > W9: In addition, the results reported for the Ego-Small dataset appear to outperform the training set itself, which should not be possible. This discrepancy likely indicates a presentation or evaluation issue and should be clarified.
> > >
> > > Q3: How the authors explain that the results for ego-small outperform the training set?
> >
> >
> > This observation stems from our adherence to the experimental setup established by GGFlow [3], which we adopted to ensure a direct and fair comparison. Specifically, this protocol involves generating a small batch of graphs (40) to compare against the test set (40). Given this limited sample size, the evaluation metric is subject to certain statistical fluctuations. Consequently, it is possible for the generated graphs to coincidentally exhibit higher similarity to the test set than the training set does within this specific setting.
> >
> > **(Response to W10): QM9 Datasets**
> > >W10: For the QM9 datasets (both with and without hydrogen), the paper does not specify which version is used (kekulized vs. aromatic bonds). This distinction is crucial, as results across these versions are not directly comparable.
> >
> > Thank you for this clarification. We were previously unaware of the distinction between the Kekulized and aromatic bond versions. Upon verification, we confirm that we utilized the **Kekulized** version, consistent with the standard setting used by the majority of baseline methods.
> >
> > ---
> > [1] Yoann Boget. Simple and critical iterative denoising: A recasting of discrete diffusion in graph generation, ICML 2025.
> >
> > [2] Itai Gat, Tal Remez, Neta Shaul, Felix Kreuk, Ricky T. Q. Chen, Gabriel Synnaeve, Yossi Adi, Yaron Lipman. Discrete flow matching, NeurIPS 2024.
> >
> > [3] Xiaoyang Hou, Tian Zhu, Milong Ren, Dongbo Bu, Xin Gao, Chunming Zhang, and Shiwei Sun. Improving graph generation with flow matching and optimal transport, 2025.

---

> ### Author Response · Authors · 2025-11-26
>
> Dear Reviewer UM94,
>
> Thank you very much for your thoughtful follow-up and for clearly outlining your derivation. We greatly appreciate your continued engagement and your interest in clarifying the relationship between vf-denoiser and rvf-denoiser.
>
> However, we respectfully posit that **Eq. 2 is likely incorrect**, as it appears to stem from a misunderstanding of the rvf-denoiser.
>
> Rvf-denoiser updates by **sampling** $x_{1|t}$ from the model output and moving towards that specific sample; it does not compute the expectation over the $x_{1|t}$ distribution during a single run. **Your derivation seems to treat the single-step realization as if it were the expectation itself**.
>
> Therefore, the correct formulation should be:
>
> $
> \mathbb{E}\_{x\_{1|t}}\left[p\_{t+h}^R(x \mid x\_t)\right] = \mathbb{E}\_{x\_{1|t}} \left[ \delta\_{x\_t}(x) + h u^{rvf}(x, x\_t) \right]
> $
>
> **In short, while rvf-denoiser and vf-denoiser are equivalent in expectation, their behavior during actual execution is invariably different.**
>
> If you have any further questions, please do not hesitate to let us know. We are more than happy to provide further clarification.
>
> Best Regards,
>
> The Authors

---

> > ### Comment · Reviewer_UM94 · 2025-11-26
> >
> > Dear Authors,
> >
> > Thank you for the rapid answer and the clarification. It is helpful to point exactly where our interpretations diverge. I fully agree that an RVF update uses a _sampled_ value $x_{1|t}$, not its expectation, and that in any single update the algorithm operates on a particular realization drawn from $p_{1|t}(\cdot \mid x_t)$.
> >
> > My derivation does not claim otherwise. Rather, the goal is to compare the **transition kernels** of RVF and VF, that is, the distributions obtained **after integrating out** the randomness introduced by sampling $x_{1|t}$.
> >
> > Formally, conditioned on both $x_t$ and the sampled $x_{1|t}$, the update distribution is:
> >
> > 1. $p^R_{t+h}(x \mid x_t, x_{1|t}) = \delta_{x_t}(x) + \lambda_t(\delta_{x_{1|t}}(x) - \delta_{x_t}(x))$
> >
> > By the law of total probabilities, we then have:
> >
> > 2. $p^R_{t+h}(x \mid x_t)  = \sum_{\hat{x}}p_{1|t}(\hat{x} | x_t) [\delta_{x_t}(x) + \lambda_t (\delta_{\hat{x}}(x) - \delta_{x_t}(x))] = E_{x_{1|t} \sim p_{1|t}(\cdot \mid x_t)}  [\delta_{x_t}(x) + \lambda_t (\delta_{x_{1|t}}(x) - \delta_{x_t}(x))]$,
> >
> > which is Eq. 2 (above), and seems correct to me.
> >
> > The expectation therefore does **not** replace the sampled value by its mean inside the algorithm. It arises naturally once we integrate over the randomness to obtain the full transition kernel. In other words, the expectation appears because we are computing the distribution of the *update*, not because we are treating the realized sample as an expectation.
> >
> > Best regards,
> >
> > Reviewer UM94

---

> ### Author Response · Authors · 2025-11-27
>
> Dear Reviewer UM94,
>
> Thank you for the continued discussion and for the rigor of your mathematical derivation. Based on your latest response, we believe the current divergence lies not in the derivation itself, **but in the definition of "equivalence" in stochastic processes**. **Your derivation correctly proves that the distributions represented by rvf-denoiser and vf-denoiser are identical in expectation, but this does not imply that rvf-denoiser and vf-denoiser are strictly identical**.
>
> **1. Clarification on Notation Definitions**
>
> We notice that the definition of $p^{R}\_{t+h}$ appears inconsistent between your two responses:
>
> - In your first response, your **Eq. 1** defined $p^{R}\_{t+h}$ as: $p^{R}\_{t+h}(x|x\_{t}) = \delta\_{x\_{t}}(x)+\lambda \_t(\delta\_{x\_{1|t}}(x)-\delta\_{x\_{t}}(x))$.
>
> - In your second response, your Eq. 2 explicitly formulated it as a conditional distribution $p^{R}\_{t+h}(x|x\_{t},x\_{1|t})$, and pointed out that integration (Expectation) is needed to obtain the marginal distribution:
>
>     $p^{R}\_{t+h}(x|x\_t) = \mathbb{E}\_{x\_{1|t}\sim p\_{1|t}(\cdot|x\_{t})}[\delta\_{x\_{t}}+\lambda \_{t}(\delta\_{x\_{1|t}}(x)-\delta\_{x\_{t}})]$
>
>
> Combining Eq. 1 and Eq. 2 from your second response, we can see that the "transition kernel" you derived is essentially:
>
> $\mathbb{E}\_{x\_{1|t}}[p^{R}\_{t+h}(x|x\_{t},x\_{1|t})]$
>
> This is completely consistent with the term $\mathbb{E}\_{x\_{1|t}}\left[p\_{t+h}^R(x \mid x\_t)\right]$ we presented in our previous response. It appears that the discrepancy effectively reduces to a notational distinction: whether the expression $\delta\_{x\_{t}}(x)+\lambda \_t(\delta\_{x\_{1|t}}(x)-\delta\_{x\_{t}}(x))$ is denoted as $p^{R}\_{t+h}(x|x\_{t})$ or $p^{R}\_{t+h}(x|x\_{t},x\_{1|t})$. In the following discussion, we will adopt the notation from your second response.
>
> **2. Equality in Expectation vs. Strict Equivalence of Operational Dynamics**
>
> You observe that the transition kernels are identical after integrating out the randomness. We must emphasize that:
>
> $\mathbb{E}\_{x\_{1|t}}[p^{R}\_{t+h}(x|x\_{t},x\_{1|t})] = p^{V}\_{t+h}(x|x\_{t},x\_{1|t}) = p^{V}\_{t+h}(x|x\_{t})$
>
> Equating these implies confusing "identical means" with the actual execution being identical. As we stated previously, this represents equality only in expectation, not the strict equivalence of the operational dynamics.
>
> **This relationship is analogous to that of Stochastic Gradient Descent (SGD) versus Gradient Descent (GD).**
>
> While $\mathbb{E}[\text{SGD Update}] = \text{GD Update}$, this does not mean SGD and GD are "strictly equivalent" algorithms. SGD introduces variance (noise) that grants it properties GD lacks, such as escaping local optima. **Similarly, our experiments suggest that the performance improvement of rvf-denoiser over vf-denoiser is also observed in practice, because rvf-denoiser reduces the amplification of noise by the scale factor, mitigating the accumulation of compounding denoising errors.**
>
> **3. Proof of Distinct Update Vectors via Variance**
>
> Below, we demonstrate that the random update vectors($p_{t+h}(x|x_{t},x_{1|t})$) represented by rvf and vf are fundamentally different.
>
> Although the first moments (expectations) are identical:
>
> $\mathbb{E}\_{x\_{1|t}}[p^{R}\_{t+h}(x|x\_{t},x\_{1|t})] = \mathbb{E}\_{x\_{1|t}}[p\_{t+h}^{V}(x|x\_{t},x\_{1|t})] = p\_{t+h}^{V}(x|x\_{t})$
>
> When we examine the second moments (variance):
>
> - Variance of vf: Since $p\_{t+h}^{V}$ acts as a constant function with respect to $x\_{1|t}$, its variance is 0:
>
>     $\text{Var}\_{x\_{1|t}}(p^{V}\_{t+h}(x|x\_{t},x\_{1|t})) = 0$
>
> - Variance of rvf: Since $p^{R}\_{t+h}$ explicitly depends on the sampled $x\_{1|t}$, its variance is strictly greater than 0:
>
>     $\text{Var}\_{x\_{1|t}}(p^{R}\_{t+h}(x|x\_{t},x\_{1|t})) > 0$
>
>
> Conclusion:
>
> $\text{Var}\_{x\_{1|t}}(p^{R}\_{t+h}(x|x\_{t},x\_{1|t})) \neq \text{Var}\_{x\_{1|t}}(p^{V}\_{t+h}(x|x\_{t},x\_{1|t}))$
>
> **This proves that the update vectors are distinct. Despite sharing the same expectation, rvf-denoiser exhibits higher variance, which directly affects how numerical errors accumulate along the trajectory.**
>
> Best Regards,
>
> The Authors

---

> > ### Comment · Reviewer_UM94 · 2025-11-27
> >
> > Dear Authors,
> >
> > Thank you very much for the time and effort spent addressing my questions.
> >
> > I apologize for confusion in notation. You are entirely correct that the consistent formulation of Equation (1) requires conditioning on $x_{1|t}$, which was only implicit in the previous notation. Since we now have a shared notation (the one used in my previous message and in your reply), I will adhere to it in what follows.
> >
> > It seems that we agree on the following points:
> >
> > a. $p^R_{t+h}(x \mid x_t, x_{1|t}) = \delta_{x_t}(x) + \lambda_t(\delta_{x_{1|t}}(x) - \delta_{x_t}(x))$
> >
> > b. $E_{x_{1|t}}[p^R_{t+h}(x|x_{t},x_{1|t})] = p^V_{t+h}(x|x_{t},x_{1|t}) = p^V_{t+h}(x|x_{t})$
> >
> > c. $Var_{x_{1|t}}$  $(p^R_{t+h}(x|x_{t},x_{1|t})) \neq Var_{x_{1|t}}(p^V_{t+h}(x|x_{t}))$
> >
> > As you noted, I also believe that the remaining disagreement does not concern the derivation itself but rather the *interpretation* of what it means for update rules to be “equivalent.”
> >
> > Where we seems to still disagree is when I complete b. with:
> >
> > $p^R_{t+h}(x|x_t) = E_{x_{1|t}}[p^R_{t+h}(x|x_t,x_{1|t})] = p^V_{t+h}(x|x_{t})$
> >
> > The disagreement likely does not follow from the derivation itself, but rather from how this identity is interpreted.
> >
> > To clarify my interpretation:
> >
> > Sampling $x_{1|t}$ is an explicit component of the RVF update. Therefore, when comparing RVF with VF as stochastic processes, the relevant objects are the **transition kernels conditioned only on $x_t$**.
> > In other words, since sampling $x_{1|t}$​ is an integral part of the RVF update mechanism, the appropriate object to compare to VF is the transition kernel
> > $p^R_{t+h}(x|x_{t})$ obtained after integrating the randomness introduced by this sampling.

---

> > > ### Comment · Reviewer_UM94 · 2025-11-27
> > >
> > > Dear Authors,
> > >
> > > Let me restate the argument in a simpler and more explicit form.
> > >
> > > Your sampling procedure consists of a two-stage update:
> > >
> > > $x_{1|t} \sim p_{1|t}(x_{1|t} | x_t)$
> > >
> > > $x_{t+h} \sim p^R_{t+h|1}(x_{t+h} | x_t, x_{1|t})$
> > >
> > > Thus, the joint distribution over $(x_{1|t}, x_{t+h})$ conditional on $x_t$
> > >  is $p_{1, t+h}(x_{1|t}, x_{t+h}|x_t) = p_{1|t}(x_{1|t} | x_t) p^R_{t+h|1}(x_{t+h} | x_t, x_{1|t})$
> > >
> > > However, we are not interested in the joint distribution but the marginal transition kernel $p^R_{t+h}(x_{t+h}|x_t)$.
> > >
> > > By marginalizing out the auxiliary variable $x_{1|t}$, we obtain
> > >
> > > $p^R_{t+h}(x_{t+h}|x_t) = \sum_{x_{1|t}} p_{1|t}(x_{1|t} | x_t) p^R_{t+h|1}(x_{t+h} | x_t, x_{1|t}) = E_{x_{1|t}} p^R_{t+h|1}(x_{t+h} | x_t, x_{1|t})$
> > >
> > > We agreed that:
> > >
> > > $E_{x_{1|t}} p^R_{t+h|1}(x_{t+h} | x_t, x_{1|t}) =  p^V_{t+h|1}(x_{t+h} | x_t)$ (Proposition 1)
> > >
> > >
> > > Therefore, $p^R_{t+h}(x_{t+h}|x_t) =  p^V_{t+h|1}(x_{t+h} | x_t)$ which shows that your two-stage sampling procedure is strictly equivalent to directly sampling from the VF transition kernel.

---

> ### Author Response · Authors · 2025-11-28
>
> Dear Reviewer UM94,
>
> Thank you very much for the continued clarification. We have carefully considered your latest explanation, and we truly appreciate the precision and effort you have put into articulating your viewpoint. We will take a bit more time to work through the details and will provide a more thorough response shortly.
>
> Best regards,
>
> The Authors

---

> ### Author Response · Authors · 2025-12-03
>
> Dear Reviewer UM94,
>
> Thank you very much for the patience and care you have devoted to this discussion. As you summarized in your penultimate message, you and we are in full agreement that, under ideal infinite-precision assumptions, the marginal transition kernels (i.e., the expectations) of rvf and vf are identical. The real point of disagreement is not this fact, but rather whether, **given this kernel equivalence, rvf remains a significant and meaningful contribution of our paper.**
>
> Previously, we tried to emphasize that "rvf and vf are not completely identical in their actual execution". However, our wording was imprecise and may have suggested that we were denying kernel equivalence; this was not our intention, and we acknowledge this mistake.
>
> Our earlier explanations may therefore have been unclear. Thanks to your insights, we have now refined our discussion for greater clarity. Below, we provide a more precise clarification.
>
> In practical DFM solving, when we take into account the backbone model’s statistical error $\delta$ and the numerical projection operator $\Pi$, the two-stage sampling scheme of rvf avoids applying the scale factor $\lambda_t$ directly to $\delta$ before projection, thereby reducing the amplification of errors and helping to mitigate the accumulation of compounding denoising errors. In fact, this intuition is now **formalized in Appendix A.6 (Numerical Stability Analysis)** of the revised paper, where we provide a rigorous derivation showing that
>
> $$
> \frac{\sup \mathcal{E}\_{\mathrm{vf}}^{\mathrm{num}}}
> {\sup \mathcal{E}\_{\mathrm{rvf}}^{\mathrm{num}}}
> \sim \frac{\lambda\_t \cdot\eta}{\epsilon},
> $$
>
> where $\eta$ is an upper bound on the model error, $\epsilon$ denotes the floating-point rounding error, and $\mathcal{E}\_{\mathrm{vf}}^{\mathrm{num}}$ and $\mathcal{E}_{\mathrm{rvf}}^{\mathrm{num}}$ denote the numerical errors of vf and rvf, respectively.
>
>
> This theoretical insight is validated by extensive ablation studies across multiple datasets under identical settings:
>
> | rvf vs vf (valid) | Step | QM9↑               | QM9H↑              | Planar↑       | SBM↑         | Tree↑        |
> | ----------------- | ---- | ------------------ | ------------------ | ------------- | ------------ | ------------ |
> | SimGFM(rvf)       | 200  | **99.8** $\pm$ 0.0 | **98.4** $\pm$ 0.1 | **100.0** $\pm$ 0.0 | **90.5** $\pm$ 4.0 | **99.5** $\pm$ 1.0 |
> | SimGFM(vf)        | 200  | 99.6 $\pm$ 0.0     | 97.7$\pm$ 0.1      | 96.0 $\pm$ 3.0      | 88.5 $\pm$ 3.0     | 95.5 $\pm$ 1.0     |
>
>
> Based on your comments, we have clarified these distinctions in the revised manuscript and have **positioned rvf more explicitly as a Monte Carlo–style DFM solver built on top of the vf formulation, with better numerical properties**, rather than a completely new theoretical framework. Your detailed feedback has substantially improved how we present the relationship between the rvf-denoiser and the vf-denoiser, and we are sincerely grateful for your valuable input.
>
> However, we would like to emphasize that **our primary contribution extends beyond the numerical improvements of rvf** and encompasses the following key aspects:
>
> * **Achieving SOTA Performance with Minimal Steps:**
>
>   We demonstrate that a simple framework can achieve state-of-the-art (SOTA) performance. Crucially, SimGFM drastically reduces inference requirements, requiring only 10–50 steps (compared to 500–1000 steps in previous methods), while still achieving new SOTA performance.
>
> * **New Insights for Graph Generation:**
>
>   As detailed in our response to Reviewer onhW, our dynamical and theoretical analyses show that valid graph structures tend to emerge as $t \to 1$, which explains the effectiveness of high-order polydec schedulers when used with vf/rvf denoisers. In contrast, Campbell's formulation deteriorates as updates shrink too rapidly. This insight clarifies why SimGFM achieves high validity with fewer denoising steps and provides valuable perspectives for future graph generation paradigms.
>
> We are deeply grateful for your thorough review of our paper and your constructive feedback, which have significantly enhanced the clarity and quality of our work. While we regret that we cannot continue our discussion, we have greatly benefited from your insights and previous exchanges.
>
> Best regards,
>
> The Authors

---

### Official Review · Reviewer_onhW · 2025-10-27

**Soundness:** 2
**Presentation:** 3
**Contribution:** 2
**Rating:** 4
**Confidence:** 5

**Summary:**

This paper propose to use another DFM formulation for simplicity and improved performance with less step numbers

**Strengths:**

The paper is well written: the formulation and notations are very clear.
The motivation is well supported, and the experiments are thorough.

**Weaknesses:**

1. Given the clarity of Figure 2, QM9 is not a proper dataset for method comparison, since almost all current methods can achieve saturating performance above 99%. It does not sufficiently support a meaningful comparison between DFM and the current state-of-the-art models.
2. Although the approach is well supported and motivated, it appears to be a relatively quick adaptation of a new DFM formulation to the current SOTA model, with only a slight modification in the reverse process: sampling  $G_1$​ instead of using the full posterior distribution. This operation is also used in other DFM formulations where $G_1$​ has to be sampled. This limits the contribution of the paper. I would encourage the authors to further clarify whether there are additional insights or meaningful changes beyond this modification to enhance the contribution of the paper.
3. The performance on a more complex molecular dataset such as MOSES is worse, and this is not explained.

**Questions:**

1. Why do the authors think that this method does not perform well on more complex molecular datasets? Have you tested it on other datasets?
2. Given the simplicity of the method, I assume there are few parameters to tune beyond the scheduler mentioned in the appendix. Could the authors confirm this?
3. Could the authors further clarify the motivation in details at the beginning of Section 3.1 for comparing with the other DFM formulation (Campbell)? Why is this formulation expected to perform better with fewer denoising steps? Potentially, are there any visualizations available to compare the actual denoising trajectories between the two formulations?

---

> ### Author Response · Authors · 2025-11-23
>
> Thank you very much for your constructive feedback and for noting the clarity of our presentation. Your comments prompted us to further articulate the methodological significance and practical impact of our approach, and we provide additional clarification below.
>
> **(Response to W1): Clarifying the Role of QM9 in Evaluating Sampling Efficiency**
> > W1: Given the clarity of Figure 2, QM9 is not a proper dataset for method comparison, since almost all current methods can achieve saturating performance above 99%. It does not sufficiently support a meaningful comparison between DFM and the current state-of-the-art models.
>
> Please note that the purpose of Figure 2 is NOT to compare our SimGFM with diffusion-based or flow-based SOTA methods on QM9, but to highlight the efficiency limitations of these models. Although all methods can achieve over 99% validity, they typically require 500–1000 denoising steps to do so, underscoring a key challenge in current research:
>
> Despite QM9's simplicity (small molecules with 2–9 nodes and 4 atom types), most existing methods still need 500–1000 steps to generate valid graphs. This shows that achieving high validity efficiently remains a significant challenge, even on simple datasets.
>
>
> In contrast, SimGFM achieves over 99% validity in just 10 steps (see Figure 1), representing a 50x to 100x reduction in inference cost compared to baselines. To our knowledge, SimGFM is the most efficient method currently available for this task.
>
> | Method | QM9 |     QM9      |
> | ------ | ---- | -------------- |
> |  | Step↓ | Valid↑          |
> |Cometh|1000|99.6 $\pm$ 0.1|
> | DisCo  | 500  | 99.3 $\pm$ 0.6       |
> | DeFoG  | 500  | 99.3 $\pm$ 0.0       |
> | SimGFM | 10   | 99.4 $\pm$ 0.1       |
> | SimGFM | 50   | 99.7 $\pm$ 0.0 |
> | SimGFM | 200  | **99.8** $\pm$ 0.0 |
>
> This efficiency gain is consistent across all tested datasets (Planar, Tree, SBM), as detailed in our experimental results below.
>
> | Method      |  | Planar    |    Planar     | Tree     |     Tree    | SBM      |   SBM       |
> | ---------- | ---- | --------- | ------- | -------- | ------- | -------- | -------- |
> |             |  Step↓    | V.U.N↑     | Ratio↓   | V.U.N↑    | Ratio↓   | V.U.N↑    | Ratio↓    |
> | DeFoG       | 50   | 95.0±3.2  | 3.2±1.1 | 73.5±9.0 | 2.5±1.0 | 86.5±5.3 | 2.2±0.3  |
> | DeFoG       | 1000 | 99.5±1.0  | 1.6±0.4 | 96.5±2.6 | 1.6±0.4 | 90.0±5.1 | 4.9±1.3  |
> | SimGFM | 20   | 94.0±4.4  | 2.3±0.6 | 88.0±4.8 | 2.5±0.9 | 82.0±4.0 | 5.6±1.1  |
> | SimGFM | 50   | 99.5±1.0  | 1.8±0.5 | 97.0±1.0 | 2.0±0.7 | 87.0±4.0 | 2.9±0.5  |
> | SimGFM | 200  | **100.0**±0.0 | 9.3±2.6 | **99.5**±1.0 | 1.5±0.2 | **90.5**±4.0 | 3.17±0.5 |
>
> **(Response to W3 & Q1): Performance on Complex Datasets**
>
> > W3: "The performance on a more complex molecular dataset such as MOSES is worse, and this is not explained."
> >
> > Q1: "Why do the authors think that this method does not perform well on more complex molecular datasets? Have you tested it on other datasets?"
>
>
> Thank you for highlighting this point. Due to limited computational resources, some results in our initial submission were not trained to full convergence. For example, the MOSES dataset typically requires around 300 epochs to achieve stable performance, but at the time of submission, we were only able to train for approximately 70 epochs. After the submission, we continued training and observed significantly improved performance, as reported below and reflected in the revised paper.
> |  Method  |    MOSES  |     MOSES     |    MOSES       |   MOSES       |  MOSES        |    MOSES      |
> | --- | -- | --- | --- | ---- | -- | -- |
> |   | Step↓ | Valid↑    | Unique↑    | Novelty↑  | Filters↑  | FCD↓      |
> | DiGress |   500   | 85.7     | **100.0** | 95.0     | 97.1     | 1.19     |
> | DisCo   |  500    | 88.3     | **100.0** | 97.7     | 95.6     | 1.44     |
> | Cometh  |   500   | 90.5     | 99.9      | 92.6     | **99.1**    | 1.27     |
> | DeFoG   | **50**  | 83.9     | 99.9      | **96.9** | 96.5     | 1.87     |
> | DeFoG   | 500  | **92.8** | 99.9      | 92.1     | 98.9     | 1.95     |
> | SimGFM  | **50**   | 88.7     | **100.0** | 95.9     | 98.4     | 0.39     |
> | SimGFM  | 200  | 90.8     | **100.0** | 94.8     | 99.0 | **0.29** |
>
> It is important to note that Validity represents only one aspect of generation quality. While our Validity score is slightly lower than DeFoG, the distributional quality of the generated molecules, as measured by FCD (Fréchet ChemNet Distance), is substantially better. FCD is widely regarded as the key metric for assessing distributional alignment between generated and real molecules. As shown above, our SimGFM achieves an FCD of 0.29, compared to DeFoG's 1.95. This indicates that SimGFM captures the underlying chemical distribution more accurately, despite a small difference in validity.
>
> We have also updated the results on other datasets, which show improved performance after sufficient training epochs, in our response to W1 and in the revised paper.

---

> ### Author Response · Authors · 2025-11-23
>
> **(Response to Q3): Clarifying the Motivation and Insights through Comparison with Campbell's DFM**
> > Q3: "Could the authors further clarify the motivation for comparing with the other DFM formulation (Campbell)? Why is this formulation expected to perform better with fewer denoising steps? Any visualizations to compare the denoising trajectories?"
>
> Thank you for raising this insightful question. In the following, we clarify our motivation and present both empirical and theoretical comparisons with Campbell's DFM to address the concerns raised.
>
> **Visualization and Observations.** We visualize the denoising trajectories of vf denoiser and rvf denoiser on four datasets (Planar, Tree, QM9, QM9H), as shown in this figure ([link to denoising trajectories visualization](https://anonymous.4open.science/r/SimGFM-F9C5/img/valid_vs_t.png)).  An interesting phenomenon emerges:
>
> *The majority of valid graphs are formed when $t$ close to 1*.
>
> When using the identity (linear) scheduler—which samples evenly spaced values of $t$ —all methods only produce valid graphs when $t$ is near 1.
>
> **This aligns with DeFoG's claim that the final stages of sampling are critical and benefit from finer-grained updates (smaller steps). Accordingly, they adopt the scheduler $f(t)=1-(1-t)^2$. Motivated by this, we extend the study to the family $f(t)=1-(1-t)^k$ and examine larger $k$.**
>
> For rvf/vf, a polydec scheduler with a large $k$  (we use $k=20$ in our experiments) allows the model to sample with values close to 1 even at early stages (i.e., for small $t$). This focuses computational resources where they are most effective, enabling the model to achieve high performance more quickly.
>
> However, as shown in our comparison on QM9 ([link](https://anonymous.4open.science/r/SimGFM-F9C5/img/validity_on_qm9.png)), **setting $k=20$ in Campbell's formulation is overly high, leading to worse performance than the $k=2$ setting adopted by DeFoG**. We provide a theoretical explanation below.
>
> **Theoretical Analysis.** Below, we provide a theoretical analysis explaining why our rvf denoiser can better leverage the polydec scheduler compared to Campbell's construction.
>
> First, Campbell's construction was developed independently of any specific scheduler. When applying it with a non-uniform scheduler, we use the *time distortion* implementation strategy proposed by DeFoG. Below, we present a simple theoretical analysis in this context:
>
> Let $\kappa_t = f_k(t)$, where $f_k(t) = 1 - (1-t)^k$ is the scheduler (or time distortion). Campbell's construction, when incorporating time distortion, can be expressed as the following inference process:
>
> $
> x_{\kappa_{t+h}} \sim x_{\kappa_t}+(\kappa_{t+h}-\kappa_t)*R
> $
>
> Applying a Taylor expansion to $f_k$ at $t$ gives:
>
> $
> f_k(t+h) = f_k(t) + f_k'(t)\,h + O(h^2).
> $
>
> Therefore:
>
> $
> \kappa_{t+h} - \kappa_t
> = f_k(t+h) - f_k(t)
> = k(1-t)^{k-1}h + O(h^2).
> $
>
> When $t$ approaches 1 and $k$ is large, the term $\kappa_{t+h} - \kappa_t$ approaches 0 extremely rapidly. Consequently, $x_{\kappa_{t+h}}$ is barely updated relative to $x_{\kappa_t}$.
>
> In contrast, the solver formula for rvf/vf can be expressed as:
>
> $
> x_{t+h}\sim \delta_{x_t}(\cdot)+h \frac{\dot{\kappa_t}}{1-\kappa_t}u
> $
>
> When $\kappa_t$ is given by the polydec scheduler, this simplifies to:
>
> $
> x_{t+h}\sim \delta_{x_t}(\cdot) + h\frac{k}{1-t}u
> $
>
> In Campbell's construction, $R$ represents the bounded model output, and the scaling factor is $(\kappa_{t+h}-\kappa_{t})$. For vf/rvf, $u$ is the bounded model output, and the scaling factor is $h\frac{k}{1-t}$. When $k$ is relatively large (e.g., 10 or 20) and $t$ approaches 1, $(\kappa_{t+h}-\kappa_{t})$ decays to $O(h^2)$, which is much smaller than the vf/rvf scaling factor $h\frac{k}{1-t}$. This strong reduction significantly limits the probability of modifying an element in Campbell's formulation. As a result, Campbell's formulation performs very few effective edits at each step, making it more vulnerable to compounding denoising errors [1].
>
> Additionally, we compare rvf, vf, and Campbell's formulation on QM9 using 10 steps with $k=20$, as shown in this figure ([link to sampling steps visualization](https://anonymous.4open.science/r/SimGFM-F9C5/img/validity_on_qm9.png)).
>
> The results indicate that **vf** (orange line) and **Campbell** (blue line) exhibit similar behavior during the initial steps. However, as the number of steps increases, vf continues to improve, while Campbell's method plateaus. This observation aligns with our theoretical analysis: at later steps (when $t \to 1$), the increment $\kappa_{t+h}-\kappa_t$ in Campbell's formulation becomes increasingly small, effectively diminishing the magnitude of its updates. As a result, Campbell's trajectory shows limited capacity for further refinement in later stages. In contrast, rvf follows a progressive pattern similar to vf but achieves consistently higher validity, suggesting that it better preserves update effectiveness throughout the sampling process.

---

> ### Author Response · Authors · 2025-11-23
>
> **(Response to Q2): Simplicity of Hyperparameter Tuning**
> > Q2: "Given the simplicity of the method, I assume there are few parameters to tune beyond the scheduler mentioned in the appendix. Could the authors confirm this?"
>
> Yes, aside from the backbone model and velocity field, the scheduler is the only hyperparameter in our SimGFM. Indeed, this simplicity is one of its key advantages.
>
> **(Response to W2): Clarifying Our Contributions**
>
> > W2: "Although the approach is well supported and motivated, it appears to be a relatively quick adaptation of a new DFM formulation to the current SOTA model, with only a slight modification in the reverse process: sampling $G_1$ instead of using the full posterior distribution. This operation is also used in other DFM formulations where $G_1$ has to be sampled. This limits the contribution of the paper. I would encourage the authors to further clarify whether there are additional insights or meaningful changes beyond this modification to enhance the contribution of the paper."
>
> We appreciate the reviewer's suggestion to clarify our contributions. While SimGFM is technically simple, we consider this simplicity a key strength. As reviewer nKK3 observed, "by introducing an elegant, simplified rate formulation, the model achieves a significant reduction in the required sampling steps while demonstrably maintaining competitive performance." Our aim was to develop a method that is both conceptually simple and practically effective. This simplicity enables SimGFM to reduce the number of sampling steps from 500–1000 to just **10–50**, while still achieving **competitive performance** across benchmarks. We believe this balance between simplicity and effectiveness is central to our contribution.
>
> Furthermore, inspired by your thought-provoking question (Q3), we present new analyses and deeper insights into our framework and its advantages over existing approaches. As summarized in our response to Q3, our study reveals an interesting observation regarding graph generation:  **In uniform denoising, valid graph structures predominantly emerge near the end of the denoising trajectory.** This finding has important implications.
>
> 1. It suggests that the generative process for graphs may benefit from sampling with values (i.e., outputs of the scheduler function $f(t)$) near 1. Such findings support previous claims in DeFoG that utilizing variable step sizes and concentrating smaller, more frequent steps toward the final stages of sampling can be beneficial.
>
> 2. It indicates the potential to employ suitable scheduler functions (such as high-order polydec functions) that enable sampling with values close to 1 during the early stages, thus significantly accelerating the generation of high-validity graphs.
>
> Our theoretical analysis further demonstrates the benefits of employing high-order (e.g., $k=20$) polydec schedulers in conjunction with vf/rvf denoisers, compared to their application within Campbell's formulation. This insight **elucidates why SimGFM is able to achieve high validity with substantially fewer denoising steps**. We believe that these findings and analyses will be valuable for future studies. The revised paper now includes these results and analyses.
>
> ------
>
> [1] Yoann Boget. Simple and critical iterative denoising: A recasting of discrete diffusion in graph generation, ICML 2025.

---

> > ### Comment · Reviewer_onhW · 2025-11-27
> >
> > Thank the authors' reply.
> >
> > While the work seems relatively straightforward to complete, I admit that the simplicity brings advantages and the experimental results are solid.
> >
> > I therefore increase my score to 6.

---

> ### Author Response · Authors · 2025-11-28
>
> Dear Reviewer onhW,
>
> Thank you for your follow-up feedback and for raising your score to 6. We appreciate your acknowledgment of the method's simplicity and the solid results.
>
> Your support is invaluable to us. Thank you once again.
>
>
> Best regards,
>
> The Authors

---

### Author Response · Authors · 2025-11-23

Dear Reviewers,

We sincerely appreciate your thoughtful and constructive comments. Over the past several days, we have refined the manuscript and prepared a focused rebuttal. The key issues we address include:

- Efficiency and clarity of the proposed formulation, including motivation and contribution (Reviewer `onhW` and `UM94`)
- Distinction between vf and rvf, and variance-based analysis (Reviewer `onhW` and `UM94`)
- Further insights into scheduler behavior, supported by denoising trajectory and a comparison with Campbell’s formulation (Reviewers `onhW` and `nKK3`)
- Additional empirical performance (Reviewer `onhW`, `UM94` and `nKK3`)

We respond to each reviewer’s comments individually, and we have updated our paper with the revisions and additional experimental results in the revised PDF.

Once again, we are grateful for your valuable feedback, which has meaningfully strengthened the clarity and depth of our work. We have made every effort to thoroughly respond to all concerns.

Thank you for your time and consideration.

Sincerely,

The Authors

---

### Meta-Review · Area_Chair_2y78 · 2025-12-26

**Summary:**

The paper proposes SimGFM, a simplified discrete flow matching framework for graph generation that empasizes a clean rate formulation and efficient sampling through a carefully designed scheduler and a stochastic denoiser variant. While the approach is technically sound and clearly presented, the decision is primarily driven by serious concerns regarding the empirical evaluation and methodological precision:

- The updated MOSES results report SOTA FCD values. However, these results are doubtful: 1) The authors do not provide SNN and Scaffold metrics, which are computed and provided jointly with FCD when using the official [MOSES benchmarking tools](https://github.com/molecularsets/moses). 2) The paper provides no description of the evaluation pipeline, and most baseline results are directly quoted from prior work. As a result, it is unclear whether the same evaluation protocols were used, making fair comparison difficult to assess. 3) MOSES contains small molecules, but the reported FCD significantly outperforms even strong SMILES-based models (see DiGress paper), which appears implausible.
- The paper's central methodological contribution, the rvf denoiser, was questioned by a reviewer as being equivalent to the standard vf-denoiser. Despite several back-and-forth discussion, this concern was not fully resolved. I believe the presentation remains imprecise, and the exact nature of the claimed novelty remains ambiguous.
- Although not explicitly raised by the reviewers, the experimental evaluation focuses primarily on small or well-saturated benchmarks (QM9 and MOSES). To substantiate claims of general effectiveness and scalability, evaluation on more structurally challenging molecular benchmarks, such as GuacaMol, is necessary.

Overall, while the paper shows some technical merit and promising ideas, the lack of rigorous, transparent and comprehensive evaluation, together with methodological ambiguities, makes it fall below the bar for acceptance. For these reasons, I recommend rejection.

**Reviewer Concerns:**

**Addressed concerns**
- The authors demonstrated that simGFM can achieve high validity on QM9 using fewer sampling steps compared to baselines.
- The authors clarified the scheduler motivation and comparison to Campbell's formulation.

**Outstanding concerns**
- The updated MOSES results appear unrealistically strong, particularly the reported FCD, and are not accompanied by the full set of standard MOSES metrics (e.g., SNN and Scaffold). The lack of evaluation details raises serious reproducibility and comparability concerns.
- In general, the manuscript does not sufficiently specify the evaluation pipeline, making it unclear whether results are directly comparable to prior work.
- Although not raised by reviewers, the evaluation largely focuses on relatively simple or saturated benchmarks. More structurally challenging benchmarks, such as GuacaMol, are absent, limiting the scope and credibility of the empirical claims.

**Reviewer Scores:**

- Reviewer onhW: 4 -> 6. This reviewer explicitly mentioned the score increase in the comment. However, they might have overlooked some potential issues in the updated results.
- Reviewer UM94: 2 -> 2. Despite extensive discussions, this reviewer still remained unconvinced regarding the claimed distinction between vf and rvf.
- Reviewer nKK3: 8 -> 8. This reviewer did not really raise any major concern.

---

### Decision · Program_Chairs · 2026-01-26

Reject